# Radionuclide-Based Imaging of Breast Cancer: State of the Art

**DOI:** 10.3390/cancers13215459

**Published:** 2021-10-30

**Authors:** Huiling Li, Zhen Liu, Lujie Yuan, Kevin Fan, Yongxue Zhang, Weibo Cai, Xiaoli Lan

**Affiliations:** 1Department of Nuclear Medicine, Union Hospital, Tongji Medical College, Huazhong University of Science and Technology, Wuhan 430022, China; d201881409@hust.edu.cn (H.L.); zhen_liu@hust.edu.cn (Z.L.); yuanlujie@qiluhospital.com (L.Y.); zhy1229@163.com (Y.Z.); 2Hubei Province Key Laboratory of Molecular Imaging, Wuhan 430022, China; 3Departments of Radiology and Medical Physics, University of Wisconsin-Madison, Madison, WI 53705, USA; kfan24@wisc.edu; 4Carbone Cancer Center, University of Wisconsin, Madison, WI 53705, USA

**Keywords:** breast cancer, radionuclide, molecular imaging, positron emission tomography (PET), single-photon emission computed tomography (SPECT)

## Abstract

**Simple Summary:**

Breast cancer is one of the most commonly diagnosed malignant tumors, possessing high incidence and mortality rates that threaten women’s health. Thus, early and effective breast cancer diagnosis is crucial for enhancing the survival rate. Radionuclide molecular imaging displays its advantages for detecting breast cancer from a functional perspective. Noninvasive visualization of biological processes with radionuclide-labeled small metabolic compounds helps elucidate the metabolic state of breast cancer, while radionuclide-labeled ligands/antibodies for receptor-targeted radionuclide molecular imaging is sensitive and specific for visualization of the overexpressed molecular markers in breast cancer. This review focuses on the most recent developments of novel radiotracers as promising tools for early breast cancer diagnosis.

**Abstract:**

Breast cancer is a malignant tumor that can affect women worldwide and endanger their health and wellbeing. Early detection of breast cancer can significantly improve the prognosis and survival rate of patients, but with traditional anatomical imagine methods, it is difficult to detect lesions before morphological changes occur. Radionuclide-based molecular imaging based on positron emission tomography (PET) and single-photon emission computed tomography (SPECT) displays its advantages for detecting breast cancer from a functional perspective. Radionuclide labeling of small metabolic compounds can be used for imaging biological processes, while radionuclide labeling of ligands/antibodies can be used for imaging receptors. Noninvasive visualization of biological processes helps elucidate the metabolic state of breast cancer, while receptor-targeted radionuclide molecular imaging is sensitive and specific for visualization of the overexpressed molecular markers in breast cancer, contributing to early diagnosis and better management of cancer patients. The rapid development of radionuclide probes aids the diagnosis of breast cancer in various aspects. These probes target metabolism, amino acid transporters, cell proliferation, hypoxia, estrogen receptor (ER), progesterone receptor (PR), human epidermal growth factor receptor 2 (HER2), gastrin-releasing peptide receptor (GRPR) and so on. This article provides an overview of the development of radionuclide molecular imaging techniques present in preclinical or clinical studies, which are used as tools for early breast cancer diagnosis.

## 1. Introduction

Breast cancer is one of the most commonly diagnosed malignant tumors, possessing high incidence and mortality rates that threaten women’s health [1]. It is a heterogeneous carcinoma, and according to different molecular subtypes, breast cancer is divided into five basic subtypes (luminal A, luminal B, HER2-enriched, basal-like, and normal-like). Different types possess different biological features, which ultimately affect prognosis, therapy response, and relapse rates. Additionally, in 2018, it was the principal cause of cancer-related deaths among females [2]. Thus, early and effective breast cancer diagnosis is crucial for enhancing the survival rate. Traditional diagnostic imaging, such as mammography, ultrasound, magnetic resonance imaging (MRI), and computed tomography (CT), is largely based on detecting changes in the anatomical structure of tumors and cannot provide information relating to the molecular characteristics of breast cancer at the early stage. Tumor invasion and metastasis are closely related to the variety of biomarkers, and although histological analysis is the primary method used to determine the expression of molecular markers, it is limited by sampling a single site at a single time point, which does not sufficiently address tumor heterogeneity [3]. In addition, this process is invasive and may cause a series of surgical complications including seroma, axillary lymphedema, and local wound infection. As a result, noninvasive molecular imaging studies have been rapidly developed in order to obtain more comprehensive biological tumor information and earlier lesion detection [4].

Molecular imaging evaluates in vivo pathophysiological processes by visualizing a specific biomarker expression—the changes in biomarkers at the cellular and molecular levels before pathological structure changes. Therefore, it can detect small lesions early and be further implemented for differential diagnosis and curative effect tracking. A molecular probe is a tool used in molecular imaging that can generate imaging signals (optical, magnetic, electrical, etc.) to achieve precise and personalized imaging. Among the various molecular probes, radionuclide probes are advantageous for clinical usage because of their 3D imaging, high sensitivity, and high specificity.

A schematic diagram of radionuclide-based imaging for breast cancer is shown in Figure 1. A radionuclide probe has three elements: targets, target agents, and radionuclides. Receptors (targets) overexpressed on tumor cells can be targeted with synthesized target ligands coupled to a chelator, often via a linker. The chelator enables labeling with radionuclides, with ligands binding their targets with high affinity and specificity. The ‘ideal’ probe for molecular imaging should have some consistent properties, including reaching the tumor site(s) rapidly, good tissue penetration, high affinity, and specificity for the tumor [3]. Since the imaging techniques and biochemical markers of breast cancer diagnosis have been reviewed elsewhere [5], this review focuses on the most recent developments of novel radiotracers as promising tools for early breast cancer diagnosis and provides an overview of novel radiotracers in different preclinical and clinical phases. These tracers can be divided into several categories, as discussed in the following sections. The graphical abstract displays an overview of several molecular pathways of breast cancer cells and targets for molecular imaging, with Table 1 providing a summary of various radiotracers for breast cancer imaging.

## 2. Imaging Biological Processes of Breast Cancer

### 2.1. Imaging Glucose Metabolism (^18^F-FDG)

The energy metabolic state of breast cancer is an important indicator for diagnosis, stratification, metastasis localization, and therapy monitoring. The tumor is characterized by augmented glycolysis even under sufficient oxygen conditions, which is known as the Warburg effect [28]. This is a common phenomenon of many types of tumors, including breast cancer, with an increased glucose metabolism usually implying a malignant phenotype and a worse prognosis. Therefore, visualizing glycolysis has become essential for tumor diagnosis. The classical PET radiotracer is 2-deoxy-2-^18^F-fluoro-D-glucose (^18^F-FDG, Figure 2A) [6]. As a glucose analog, ^18^F-FDG is imported into cells by glucose transporters and subsequently phosphorylated by hexokinase, metabolizing to ^18^F-FDG-6-phosphate. However, unlike glucose, ^18^F-FDG-6-phosphate cannot metabolize further and is trapped in the cell, meaning ^18^F-FDG imaging reflects the cellular glycolysis because of a proportional ratio between glucose metabolism and ^18^F-FDG metabolism, known as the lumped constant [29,30]. Compared to normal cells, the proliferation rate of breast cancer cells is faster, in addition to the aerobic and anaerobic glycolysis of glucose. Additionally, the carbohydrate utilization rate is increased in breast cancer cells, with an increased hexokinase concentration present in the cytoplasm [31]. Thus, most malignant breast tumors exhibit high ^18^F-FDG uptake [32].

^18^F-FDG imaging can be used for breast cancer staging, molecular subtype determination, and treatment monitoring. The European Society for Medical Oncology guidelines [34] recommend using ^18^F-FDG in early-stage breast cancer when conventional examination methods are inconclusive. A study held in Japan showed that the sensitivity and positive predictive value (PPV) of ^18^F-FDG PET screening for breast cancer were 83.9% and 41.7%, respectively [35]. However, Bertagna et al. [36] indicated SUV alone should not be used to differentiate between malignant and benign incidentalomas because there is overlap between SUVs, drawing the conclusion that ^18^F-FDG PET/CT is not routinely recommended for the initial diagnosis of primary breast cancer [37,38].

In relation to staging, one study performed a comprehensive literature review assessing indications for FDG-PET/CT in breast cancer and indicated that FDG-PET/CT is useful for staging patients with breast cancer independently of tumor phenotype and regardless of tumor grade [39]. In a recent study, Yararbas et al. [33] concluded that ^18^F-FDG made a significant contribution to the accurate staging of breast cancer, starting from stage IIA (Figure 2B,C). ^18^F-FDG can detect metastases to mediastinal, axial, and internal mammary nodes, and a meta-analysis revealed ^18^F-FDG PET/MRI demonstrates high diagnostic value in the TNM staging in breast cancer patients and can serve as a promising imaging biomarker for future evaluation of the TNM stage of breast cancer [40]. Han et al. [41] performed a systematic review and meta-analysis to evaluate the impact of ^18^F-FDG on staging and management as an initial staging modality of breast cancer. The results suggested that routine clinical use of ^18^F-FDG PET, PET/CT, or PET/MRI imaging leads to significant modification of the initial staging in newly diagnosed breast cancer patients.

Additionally, in a study conducted by Arslan et al. [42], ^18^F-FDG was used to evaluate the molecular subtypes and clincopathological features of primary breast cancer, revealing an association between a high maximum SUV (SUVmax) and more aggressive behavior. Moreover, ^18^F-FDG uptake is related to molecular subtypes luminal A, luminal B, human epidermal growth factor receptor 2 (HER2)-positive, and triple-negative [43]. In 2015, Kitajima et al. [44] reported that triple-negative and HER2-positive breast cancers have a higher SUVmax, while the luminal A subtype has a lower SUVmax.

For treatment monitoring, ^18^F-FDG has been widely applied in the management of neoadjuvant chemotherapy (NAC) for locally advanced breast cancer patients. It can effectively monitor the therapeutic response and improve patients’ quality of life [45]. A meta-analysis in 2013 concluded that ^18^F-FDG imaging can accurately predict NAC’s curative effect on breast cancer in the early to mid stage, which has moderately high sensitivity and specificity [46]. Han et al. [47] also reported that ^18^F-FDG provided significant predictive value for the evaluation of responses to NAC in breast cancer patients and might guide rational management. Caldarella et al. [48] performed a meta-analysis of 8 studies with 873 suspected breast cancer cases and came to a similar conclusion. In an international multicenter prospective study, Gebhart et al. [49] evaluated the efficacy of lapatinib and trastuzumab on developing breast cancer patients. ^18^F-FDG imaging was performed in the baseline period before treatment, and on the second and sixth weeks after treatment. The results showed a correlation between ^18^F-FDG uptake at 2 weeks and 6 weeks after treatment, meaning patients who are effective in the second week after targeted therapy are usually effective in the sixth week after treatment. This study indicated that ^18^F-FDG PET imaging can predict the efficacy of targeted therapy at the early stage, without waiting for the middle stage or the end of treatment. Therefore, ^18^F-FDG PET imaging was included in future studies as an essential biological detection method, providing a reference for the clinical decision of NAC and endocrine therapy. However, the specific threshold value [50] and the definition of good histopathologic response varies [51], and the optimal timing of interim PET is unclear [52]. These disparities need to be standardized.

It is important to note that ^18^F-FDG is not supposed to diagnose inflammatory breast cancer [34,53] due to its uptake within inflammatory cells complicating the interpretation of imaging results [54]. Roughly 25% of ^18^F-FDG uptake concentrates in granulation or fibrous tissues [55], resulting in confusion between residual tumor tissues and changes occurring after treatment [56]. In the early period of post-therapy, there is an inflammatory response in tumor cells and the surrounding normal cells, meaning ^18^F-FDG uptake tentatively increases in both types of cells, which ultimately decreases specificity. ^18^F-FDG uptake can only reflect the glucose metabolism of tumor cells without providing the distribution of receptors or tumor proliferation. Other benign lesions with high ^18^F-FDG uptake will also affect diagnostic accuracies, such as infection, fibroadenoma, and ductal adenoma. Furthermore, the sensitivity is relatively low for the diagnosis of sub-millimeter tumors due to the limited spatial resolution [48].

### 2.2. Imaging Amino Acid Metabolism

The uncontrolled proliferation of tumor cells enhances cellular biosynthesis and division, including the processes of glycolysis, nucleotide synthesis, protein synthesis, and lipid synthesis. The enhanced protein synthesis requires increased amino acid (AA) intake. The most commonly used amino acid PET imaging agent is L-methyl-^11^C-methionine (^11^C-MET) [7]. Methionine, an essential amino acid of the human body, is crucial for tumor growth [57], playing a vital role in protein synthesis and methylation. ^11^C-MET has been applied to measure methionine accumulation in breast cancer patients, with high uptake of ^11^C-MET correlating with a high S-phase fraction of breast cancers, measured by flow cytometry [58]. This result indicates that ^11^C-MET uptake might relate to the proliferation rate of breast cancer.

^11^C-MET PET imaging can evaluate the early curative effect of advanced breast cancer. In one study, Huovinen et al. [59] studied eight patients with breast cancer metastases using ^11^C-MET, evaluating the effect of treatment. They found that ^11^C-MET uptake decreased in metastases that responded to treatment, whereas it increased when subsequently developed metastases occurred during treatment. Jansson et al. [60] established similar findings in 1995, studying 16 patients with locally advanced, recurrent, or metastatic breast cancer using the radiotracer ^11^C-MET. Lindholm et al. [7] in 2009 assessed the early response to therapy of metastatic breast cancers using ^11^C-MET PET imaging. In their study, 13 advanced breast cancer patients underwent ^11^C-MET PET imaging both before and after the first period of polychemotherapy, or after the first month of hormone therapy, or low-dose weekly cytostatics. The curative effect of treatment was evaluated by comparing SUV changes before and after treatment. In responders, the SUVs decreased significantly after treatment. However, the SUVs declined mildly, remained stable, or increased uptake in non-responders, with these results confirming previous conclusions. In another study, Inoue et al. [61] compared the ability of ^18^F-FDG and ^11^C-MET to detect residual or recurrent tumors. They studied 24 patients with 34 lesions using ^18^F-FDG and ^11^C-MET, finding equal effectivity among these two approaches, with the uptake of ^18^F-FDG being somewhat higher than that of ^11^C-MET. However, for small tumors, both showed a limited diagnostic value.

The physiological intake in the pancreas, liver, bone marrow, and other normal tissues limited the further clinical application of ^11^C-MET in breast cancer for therapeutic effect evaluation [62]. Furthermore, the short half-life of ^11^C leads to rapid metabolism and places time constraints on image acquisition, which can reduce the image quality and hinder ^11^C-MET PET’s application in tumor imaging [63,64].

Fluorine-18-labeled AAs have longer half-lives than ^11^C-MET, enabling better detection of tumor AA metabolism and amino acid transporters [65,66,67,68]. Ideally, an ^18^F-labeled AA PET probe should conform to some specific conditions. Firstly, it should be transported into tumor cells rapidly with a high uptake rate. Secondly, it should stay in the cell for a certain amount of time. Thirdly, the blood clearance rate should be high. Fourthly, it should not combine with the non-protein or inflammatory tissues. Lastly, the labeling method should be relatively simple and practical [69,70]. Many commonly used clinical fluorine-18-labeled AAs have been developed and meet the above conditions, including^18^F-FDOPA for gliomas [71,72,73] and neuroendocrine tumors [74,75,76,77,78]; ^18^F-OMFD [79] and ^18^F-FET [80,81,82] for brain tumors; ^18^F-FAMT for brain, oral cavity, and non-small cell lung cancers [83]; ^18^F-FACPC [84,85,86] and ^18^F-FACBC [87,88,89,90,91,92,93,94] for prostate cancers; and ^18^F-FGln [95], ^18^F-FASu [96,97], and ^18^F-fluciclovine [8,98,99,100] for breast cancers.

^18^F-(2S, 4R) 4-fluoroglutamine (^18^F-FGln) could be used to measure the glutamine pool size of TNBC cells [95]. Many aggressive tumors utilize glutamine for survival through glutaminolysis. High glutaminase (GLS) activity leads to a small glutamine pool size, whereas GLS inhibition markedly increases the glutamine pool size. The change in the glutamine pool size may reveal a drug’s pharmacodynamic effect on the glutaminolysis pathway. ^18^F-5-fluoro-aminosuberic acid (^18^F-FASu) may be a valuable target for monitoring the diagnosis and therapeutic effect of breast cancers [96]. As an endogenous cellular antioxidant, glutathione plays an important role in coping with oxidative stress (OS) by neutralizing free radicals. Cysteine/glutamate transporter activity represents glutathione biosynthesis in the process of responding to oxidative stress and is expressed relatively low in most normal tissues. While cells are under OS, they are upregulated for the antioxidant response. ^18^F-FASu was developed as a potentially useful PET imaging tracer that targets the cysteine/glutamate transporter and might be more sensitive to certain tumors compared to ^18^F-FDG [97].

^18^F-labeled 1-amino-3-fluorocyclobutane-1-carboxylic acid (^18^F-fluciclovine, ^18^F-FACBC) is a leucine analog radiotracer that depicts amino acid transport into cells [8]. Its uptake in malignant breast cancers was higher than benign lesions and normal surrounding tissues. ^18^F-fluciclovine PET/CT imaging provides a new method for visualizing invasive lobular and invasive ductal breast cancer. Ulaner et al. [98] found that ^18^F-fluciclovine can also readily detect bone, lung, brain, and axillary nodal metastases, but its ability to detect liver metastases was limited due to the prominent physiologic uptake in the liver parenchyma. Tade et al. [99] studied the correlation of ^18^F-fluciclovine uptake with the histologic and immunohistochemical features in breast cancer, finding that the radiotracer uptake in the triple-negative and Nottingham grade 3 subtypes was the highest.

### 2.3. Imaging Cell Proliferation

Increased cell proliferation is one of the essential characteristics associated with tumor biological behavior. The prognosis and aggressiveness of a tumor can be understood by detecting the state of tumor proliferation. Generally, most studies have focused on imaging the thymidine salvage pathway during DNA synthesis because thymidine is the only pyrimidine or purine base incorporated into DNA rather than RNA [101,102]. This process uses thymidine analog radionuclide probes, including ^11^C-thymidine, 3′-deoxy-3-^18^F-fluorothymidine (^18^F-FLT), and 1-(2′-deoxy-2′-fluoro-1-β-D-arabinofuranosyl)-thymine (FMAU). ^11^C-thymidine, one of the first thymidine proliferation probes, measures the different protein synthesis rates between normal and tumor cells, with studies showing a connection between ^11^C-thymidine uptake and the S-phase fraction in cancer [58,60]. However, the short half-life of ^11^C, rapid catabolism after injection, complicated radiosynthesis and modeling analysis, and low tumor uptake indicate that it is not an ideal radiotracer for imaging tumors’ proliferative status, and it has generally been abandoned [103].

Currently, the most promising radiotracer for cell proliferation is ^18^F-FLT (Figure 3) [9]. ^18^F-FLT is phosphorylated by thymidine kinase-1 (TK-1) but cannot further participate in DNA synthesis and is trapped within the tumor cells due to the lack of the 3′-hydroxy group. ^18^F-FLT uptake has been reported to significantly relate to Ki-67 expression in breast cancer [104]. By studying this radiotracer’s uptake and kinetics, the status of cell proliferation and DNA synthesis can be visualized in vivo. Unlike ^18^F-FDG, ^18^F-FLT is not concentrated in inflammatory tissues, avoiding false positives [105]. However, the tumor uptake rate and tumor-to-normal tissue contrast of ^18^F-FLT are lower than those of ^18^F-FDG, as proliferative cells grow asynchronously, whereas glucose metabolism is associated with many factors, not only cell proliferation. The lower normal tissue uptake and limited accumulation within inflamed tissues resulted in a higher contrast of ^18^F-FLT in tumors. In addition, the physiological concentration of ^18^F-FLT in the hyperproliferative tissues, such as the liver and bone marrow, may limit its utility in tumor imaging to some degree [9]. The high background uptake in the liver for glucuronidation may limit its clinical application for liver metastases [106]. No studies have reported ^18^F-FLT uptake in benign lesions related to high proliferation rates.

Several experimental studies have used ^18^F-FLT to evaluate the early response to breast cancer treatment. In one study, Ellis et al. [107] showed that changes in cell proliferation status after chemotherapy or endocrine therapy were associated with prognosis in breast cancer. The evaluation of cell proliferation status is essential to determine the efficacy and prognosis of breast cancer patients. A study from the University of California also showed that ^18^F-FLT uptake can predict changes in tumor proliferation after one course of treatment with cytotoxic chemotherapy for breast cancer [108]. Pio et al. prospectively studied 14 patients with newly diagnosed early or advanced breast cancer, implementing a new pharmacologic treatment schedule. Patients were scanned with ^18^F-FLT three times: before starting the new regimen, 2 weeks after the first course of treatment, and at the end of chemotherapy. After the first cycle of treatment, the change in SUV values was significantly associated with the tumor marker CA27-29 of breast cancer. There was also a high correlation between tracer uptake and tumor size changes measured by CT, further revealing the usefulness of ^18^F-FLT when monitoring the efficacy of chemotherapy for breast cancer patients. Kenny et al. [109] found that ^18^F-FLT uptake is significantly different between responders and non-responders 6–12 days after treatment. The ^18^F-FLT response is usually earlier than the changes in the tumor diameter. These results demonstrated the potential utility of ^18^F-FLT in determining a positive response as early as 1 week after chemotherapy, which is superior to the 5 weeks posttreatment of ^18^F-FDG [108].

FMAU is another fluorine-18 labeled thymidine analog that can be easily incorporated into DNA after phosphorylation by TK-1. This occurs because it contains the 3′-hydroxy group and offers a direct method for DNA synthesis. Sun et al. [110] reported good tumor-to-normal tissue ratios of FMAU after studying 14 cases. In breast cancer, the average SUV value was 2.17, which can clearly show lesion areas. Compared with ^18^F-FLT, 5–10 times lower uptake of FMAU was noted in tumors such as TNBC. In addition, the bone marrow uptake level was also lower, but physiologic liver uptake was also observed because FMAU is a substrate of the mitochondrial enzyme thymidine kinase-2 (TK-2) with low specificity for TK-1. It can participate in mitochondrial DNA synthesis, leading to higher physiologic uptake in normal tissues [110]. Therefore, FMAU is less desirable than ^18^F-FLT for imaging the proliferative status of tumors.

### 2.4. Imaging Hypoxia

Hypoxia is an independent negative prognostic factor that contributes to tumor progression, invasion, and metastasis. Patients with hypoxic cancers often have a poorer prognosis, low chemotherapy/radiotherapy efficiency, and a lower survival rate. Given its prominent role in oncology, affecting prognosis and treatment planning, noninvasive and accurate monitoring of tumor hypoxia is highly clinically significant.

Tumor hypoxia is associated with decreased oxygen partial pressure compared to the surrounding normal tissue. A variety of techniques have been developed to monitor hypoxia, including oxygen electrodes, near-infrared spectroscopy, electron paramagnetic resonance spectroscopy, blood or tissue oxygen level-dependent MRI, SPECT, and PET. Among these techniques, hypoxia PET scanning provides noninvasive 3D imaging and quantifies intratumor oxygen levels through various hypoxia probes.

The cell’s response to hypoxia is principally controlled by hypoxia-inducible factors (HIF). Karakashev et al. [111] showed that hypoxia and its biological marker HIF are connected with a tumor’s cell proliferation, metastasis, recurrence, and drug resistance to treatment. HIF-1 therein plays the most crucial role in the cellular response to hypoxia. The findings by Generali et al. [112] in a 2011 study show that the combined treatment of breast cancer with the new adjuvant chemotherapy drug letrozole and cyclophosphamide increased the HIF-1 level, thereby increasing the antagonistic response of treatment. These results indicate that hypoxia may be associated with ineffective endocrine therapy for some breast cancers.

^18^F-fluoromisonidazole (^18^F-FMISO, Figure 4A) is one of the most used PET hypoxia imaging probes [10]. The sufficiently lipophilic nature ensures easy facilitation to penetrate the cell membrane, enter the cell, and be uniformly distributed within the tissue. ^18^F-FMISO has been demonstrated in several tumor types including gliomas [113] and breast cancers [114]. A study led by Cheng et al. [114] examined ER-positive breast cancer patients using ^18^F-FMISO PET/CT for baseline and post-endocrine therapy imaging to predict treatment outcomes (Figure 4B). Following the analysis of 33 lesions within 16 ER-positive breast cancer patients, the predicted values of tumor metastasis and partial remission were up to 88% and 100% with the application of a 4 h tumor-to-background ratio cutoff of ≥1.2.

Many studies confirmed that ^18^F-FMISO could evaluate tumor hypoxia in vivo, and it became the lead candidate to assess hypoxia with PET. However, ^18^F-FMISO has not been used for routine clinical diagnostics because of its slow pharmacokinetics: a slow clearance from blood and normal tissues leads to a modest hypoxic-to-normoxic ratio and limited contrast images. The contrast is limited in defining hypoxic tumor imaging by using a tumor/background ratio of ≥1.2, which makes visual examination of hypoxic areas difficult and hinders its diagnostic application in clinical oncology. Therefore, hypoxia tracers with improved pharmacokinetic properties that are more amenable to clinical use are in high demand, especially ones with enhanced clearance from normoxic tissues.

Fleming et al. [10] summarized the available hypoxia probes in 2015 and presented the properties of ideal hypoxia tracers. These characteristics include a high specificity to hypoxia, a good lipid–water distribution ratio, high in vivo stability, accessibility to synthesis, and efficiency to various tumor types. Other new fluorine-18 labeling nitroimidazoles radioactive drugs have been studied intensively. For example, ^18^F-fluoroazomycin-arabinofuranoside (^18^F-FAZA, Figure 4) [115], with good serum metabolism, may have applications in breast cancer’s therapeutic effect evaluation in the near future. Compared with ^18^F-FMISO, ^18^F-FAZA is more hydrophilic and has faster clearance kinetics. Therefore, its tumor-to-reference tissue ratios, specifically the hypoxia-to-normoxia contrast, are improved. The radiolabeled probe ^18^F-FAZA has been successfully applied in gliomas, lymphomas, etc. [116]. Although ^18^F-FAZA is not currently widely used, it is gradually becoming popular for tumor hypoxia PET imaging.

Other hypoxia probes include ^18^F-fluoroerythronitroimidazole (^18^F-FETNIM) [117], ^18^F-1-(2-1-(1H-methyl) ethoxy)-methyl-2-nitroimidazole (^18^F-RP-170) [118], and ^18^F-3-fluoro-2-(4-((2-nitro-1H-imidazol-1-yl)methyl)-1H-1,2,3-triazol-1-yl)propan-1-ol (^18^F-HX4) [119]. Furthermore, ^64^Cu-ATSM, a hypoxia probe based on diacetyl-bis(N4-methylthiosemicarbazone) ligands, is characterized by high membrane permeability on the account of its lipophilicity and lower molecular weight, which can rapidly diffuse into cells. Compared to other tracers, ^64^Cu-ATSM has several advantages for tumor hypoxia imaging, including a simpler radiolabeling method, faster clearance rates, shortened intervals from injection to imaging, and a higher hypoxic-to-normoxic contrast. The effectiveness of ^64^Cu-ATSM has been verified in lung carcinomas [120], cervical cancers [121], rectal tumors [122], and gliomas [123]. However, its effectiveness for breast cancer diagnosis is yet to be evaluated.

### 2.5. Imaging Cellular Transmembrane Electrical Potential

Tumor cells are characterized by greater energy-dependent metabolism, a higher proliferation rate, and increased resistance against apoptosis. One of the main contributors is the significantly increased mitochondrial membrane potential (MMP) in tumor cells. The increased cellular transmembrane electrical potential leads to increased uptake of cell-permeant cationic compounds. With hints of this mechanism, lipophilic cation analogs are labeled with radionuclides for breast scintigraphy, of which ^99m^Tc-sestamibi and ^99m^Tc-tetrofosmin are the two most widely used radiotracers.

^99m^Tc-methoxy isobutyl isonitrile, also known as ^99m^Tc-sestamibi (^99m^Tc-MIBI), is a small lipophilic cationic radiopharmaceutical. ^99m^Tc-tetrofosmin (1,2-bis bis(2-ethoxy-ethylphosphine)ethane) is a lipophilic diphosphine compound. These two radiotracers concentrate most in mitochondria and are taken as probes for MMP, denoting the cellular transmembrane electrical potentials. They are accumulated in various neoplasms and the myocardium. Therefore, they are used as tumor imaging probes as well as myocardial perfusion imaging agents. Typically, they are commonly used to evaluate malignant pathology within breast tissues. They can not only reveal the lesion site but also reflect the specific biological and functional characteristics of the lesion, including perfusion, proliferation potential, metabolic activity, and receptor status. The early tracer uptake mechanism is driven by a negative transmembrane potential depending on mitochondria, which is related to an increased energy-dependent metabolism and cell proliferation. Due to the biochemical and physiological characteristics of malignant tumors, the increase in neovascularization, blood perfusion, cell proliferation, and metabolism leads to negative transmembrane potential enlargement. Therefore, ^99m^Tc-MIBI and ^99m^Tc-tetrofosmin imaging showed an obvious radioactive concentration in tumor lesions.

Notably, ^99m^Tc-MIBI was approved by the Food and Drug Administration in June 1997 and is the first radiopharmaceutical used for radionuclide breast imaging, which promoted the research of scintimammography in breast cancer detection [124]. Scintimammography is considered as a complementary diagnostic procedure to mammography when the breasts are mammographically dense or mammography is doubtful, inadequate, or indeterminate. Khalkhali et al. [125] reported a multicenter study performed in 558 women that prospectively enrolled 580 abnormal breasts, including 276 dense breasts, and concluded that the diagnostic accuracy of ^99m^Tc-MIBI breast scintigraphy is not affected by breast density. Lumachi et al. [126] evaluated the effectiveness of ^99m^Tc-MIBI scintimammography and mammography in 87 premenopausal patients with suspicious breast lesions smaller than 2 cm. The sensitivity, specificity, positive predictive value, negative predictive value, and diagnostic rate were 81% vs. 81%, 93% vs. 60%, 98% vs. 91%, 50% vs. 39%, and 83% vs. 77%, respectively. In particular, scintimammography is more specific than mammography for patients with architectural distortions of the breast resulting from previous breast surgery, radiation therapy, chemotherapy, or biopsy [127].

Conventional scintimammography uses the planar acquisition method, which has a low sensitivity to lesions smaller than 10 mm. Early clinical studies reported that the sensitivity and accuracy of SPECT are higher than those of planar imaging in the detection of both small non-palpable primary breast cancer and axillary lymph node metastasis [128,129]. Using a pinhole collimator (pinhole SPECT) can further improve the performance of SPECT. The better spatial resolution allows it to determine the number of involved nodes, thus guiding the physician more accurately in fine-needle aspiration [130]. The development of hybrid SPECT/CT devices, especially high-resolution specific breast cameras, has significantly improved the detection rate of subcentimeter malignant lesions. A prospective study reported that ^99m^Tc-tetrofosmin SPECT/CT proved a useful diagnostic tool in the detection of both residual breast tumors and axillary lymph node metastases following neoadjuvant therapy, and that it may guide the surgeon to the most appropriate breast surgical treatment and select the most suitable axillary lymph node sampling [131]. With the implementation of a high-resolution dedicated breast camera (DBC), planar scintimammography was found to have much improved sensitivity in monitoring the neoadjuvant chemo/hormonotherapy response in locally advanced primary breast cancer, especially in detecting microscopic residual tumor foci [132]. Planar scintimammography equipped with a high-resolution DBC showed technical advantages and better clinical performance than SPECT in the detection of subcentrimetric carcinoma and the assessment of multifocal/multicentric disease [133,134]. Breast-specific gamma imaging (BSGI), a high-resolution radionuclide imaging approach, uses a small field-of-view gamma camera to visualize breast tissues that are confined to the breast region. BSGI with ^99m^Tc-tetrofosmin has proved to be a highly sensitive diagnostic tool in the detection of ductal carcinoma in situ (DCIS) independent of histologic subtype, and it has demonstrated slightly higher sensitivity than mammography and a better assessment of the local disease extent [135]. Moreover, BSGI proved a highly sensitive diagnostic tool with a high specificity even in small size carcinoma detection, which increased the sensitivity and specificity of mammography [136]. The study by Rhodes et al. [137] on the assessment of the diagnostic performance of BSGI or molecular breast imaging (MBI) at a reduced radiation dose in women with dense breasts reported that the supplemental cancer detection rate of MBI was 8.8 cancers per 1000 women (increased from 3.2 to 12.0) when added to mammography. Thus, MBI offers a supplemental screening option in women with mammographically dense breast tissue [138]. From a meta-analysis, Liu et al. [139] retrospectively analyzed data from 177 women with BI-RADS 4 category lesions that had undergone BSGI and were originally detected via ultrasound and/or mammography, and compared the relative diagnostic utility of these three approaches. The results indicated that BSGI is highly sensitive for the detection of BI-RADS 4 category lesions, achieving good positive/negative predictive values, and is superior to ultrasound and mammography for invasive ductal carcinomas. A recent retrospective review summarized that MBI with ^99m^Tc-tetrofosmin using a high-resolution, solid-state dedicated breast camera was a highly accurate diagnostic tool in predicting the complete tumor response to neoadjuvant therapy and residual tumor extent [140].

One has to be cautious when interpreting the images obtained from ^99m^Tc-MIBI or ^99m^Tc-tetrofosmin imaging because these two radiotracers are also found to be related to multidrug resistance. Researches demonstrated that ^99m^Tc-MIBI and ^99m^Tc-tetrofosmin are transmembrane P-glycoprotein (Pgp) transport substrates which are responsible for multidrug resistance [141,142]. Human Pgp is encoded by the multidrug resistance gene (MDR1) and acts as an energy-dependent drug efflux pump [143]. The tracer clearance reflects the activity of Pgp. The efflux rate for ^99m^Tc-MIBI of breast cancer tumors with high Pgp expression was 2.7-fold higher than that of tumors with low or no Pgp expression [144]. Therefore, ^99m^Tc-MIBI uptake can be significantly decreased in tumor cells overexpressing MDR1, complicating the interpretation of the imaging results. The effect of MMP on tumor cell ^99m^Tc-MIBI and ^99m^Tc-tetrofosmin uptake is confounded in the presence of high MDR1 protein expression that contributes to the tracer efflux [145]. Therefore, for better interpretation of images, it is useful to explore the history of drug resistance in patients and measure the expression levels of Pgp in patients. On the other side, scintimammography using these two radiotracers can provide functional imaging to evaluate and predict the tumor response to chemotherapy for breast carcinoma [146].

## 3. Imaging Receptors in Breast Cancer

### 3.1. Targeting Estrogen Receptor (ER)

Estrogen has a significant influence on the growth, differentiation, and function within many tissues, including the breasts [147]. Tumor growth is strictly regulated by steroid hormones such as estrogen and peptide growth factors such as HER2 [148]. About 70% of breast cancers are ER positive [149], which relates to abnormal ER signaling pathways, namely, estrogen-dependent breast cancer [150]. There is little to no expression of ER in normal cells [151], but it is highly expressed in malignant breast cells [152]. Currently, endocrine therapy selection among breast cancer patients is mainly based on the expression of ER, progesterone receptor (PR), and HER2. ER and PR play an essential role in the prognosis of breast cancer. ER- or PR-positive breast cancers are usually less aggressive and have a more favorable prognosis due to their positive response to anti-hormone therapy [153]. Compared with primary breast cancer, ER, PR, and HER2 expression changes in 20% of tumors after metastasis [154].

Immunohistochemistry staining is considered the ‘gold standard’ for assessing these receptors [155]. However, due to tumor heterogeneity and phenotypic receptor changes over time, biopsy occasionally does not necessarily capture useful information [156]. Moreover, biopsy samples are complicated by decalcification after advanced breast cancer metastases to bone. Multiple tissue sampling is required, which is not in line with clinical practice. Therefore, noninvasive detection can accurately evaluate the expression of these receptors and can predict the therapeutic response more reliably. In recent years, the application of ^18^F-labeled ER PET probes has become a research hotspot in the early diagnosis of breast cancer.

ER imaging has been extensively studied in recent years. The probes are compounds obtained from ER ligands (especially endogenous estradiol, E_2_), appropriately modified and labeled with radionuclides. ER has two specific intracellular subtypes, ERα and ERβ [157], both with different tissue distributions and biological roles [158]. In many types of breast cancer, the predominant ER expression is ERα, which mainly promotes cell proliferation and is closely associated with breast cancer. Various radioactive probes with high affinity and specificity of ER have been developed successively. The latest ER_α_-targeting PET probes are fluorine-18-labeled E_2_, and 16α-^18^F-17β-estradiol (^18^F-FES) (Figure 5A) [159], which has been the most successful in clinical trials. ^18^F-FES is an E_2_ analog, and its binding specificity is similar to E_2,_ with the affinity for ERα being even slightly higher than that of E_2_ [160]. ^18^F-FES can be specifically combined with ER after its injection into the body, which can dynamically and quantitatively reflect the expression level and distribution of ERα [161,162]. It can not only reveal the primary and metastatic lesions of breast cancer but also show normal, benign, and malignant uterine fibroids [163,164]. The tissue uptake level of ^18^F-FES correlated well with the ER expression levels obtained by immunohistochemical detection of fresh tissues [165]. Previous clinical studies reported that the sensitivity and specificity of ^18^F-FES for tumor detection were 69–100% and 80–100%, respectively [11]. By measuring the SUV of tumor ER expression, ^18^F-FES can predict whether tumors are effective for endocrine therapy drugs such as selective ER modulators or aromatase inhibitors [166]. For responders, tumor imaging has a significantly higher SUV, while negative patients with low or no ER expression are unlikely to benefit from endocrine therapy. Therefore, ^18^F-FES PET imaging can help patients avoid the adverse reactions caused by unnecessary treatment.

One disadvantage of ^18^F-FES is the lack of precise SUV thresholds when distinguishing specific uptake from nonspecific uptake. Further enhancing ^18^F-FES’s ability to predict therapeutic outcomes and allow application to more multicenter therapeutic evaluation studies is essential. Moreover, ^18^F-FES is mainly metabolized by the hepatobiliary system and excreted through the intestines [167], with rapid blood clearance resulting in lower tumor uptake. E_2_ has no selectivity for ERα and ERβ. ^18^F-FES is still less selective, which also limits the specificity of ^18^F-FES PET imaging. Over the past 20 years, researchers have synthesized a series of ^18^F-FES derivatives to improve the metabolic stability and affinity of ^18^F-FES—the modified sites include C-16α, C-11β, C-7α, and C-17α [168]. For example, Paquette et al. [169] reported that 4-fluoro-11β-methoxy-16α-^18^F-fluoroestradiol (4FM-^18^F-FES, Figure 5A) achieved significantly higher tumor uptake and tumor-to-background contrast compared to ^18^F-FES, making it a promising probe for ER imaging (Figure 5C). However, as with many other published FES derivatives, 4FM-^18^F-FES still fails to effectively address the metabolism problem of rapid blood clearance in vivo. Recently, Xu et al. [170] developed a novel probe, 1(2-(2-(2-[^18^F]fluoroethoxy)ethoxy)ethyl)-1H-1,2,3 triazole estradiol (^18^F-FETE). It showed a high radiochemical yield, purity, molar activity, and good in vitro stability. 17α ethinyl estradiol, part of ^18^F-FETE, possesses a high affinity with ER, which is 1.9 times that of E_2_ [171]. In vivo bio-evaluation revealed that ^18^F-FETE had high uptake in the uterus of normal mice and the tumors of ER-positive MCF-7-bearing mice, with effective inhibition occurring as well. However, in ER-negative MDA-MB-231-bearing mice, tumor uptake was relatively low. Compared to the tumor uptake in ER-positive mice of ^18^F-FES, ^18^F-FETE might be a promising probe in ER-positive breast cancer PET imaging. The addition of polyethylene glycol (PEG) moieties can theoretically lower lipophilicity, decrease liver metabolism, and prolong its lifetime within the blood [12]. Although ^18^F-FETE has a lower log P value than other estrogen radiotracers and lower lipophilicity than ^18^F-FES, the experimental result was unsatisfactory. The rapid metabolism problem in vivo was still present. In the future, this research team plans to add different PEG modifications to 17α-ethinyl estradiol.
Figure 5(**A**) chemical structure of ERα receptor probes marked by ^18^F. Estradiol (E2), ^18^F-FES, and 4FM-^18^F-FES; (**B**) image of the bone scan (left) and ^18^F-FES PET (right) of a patient after multiple lines of anti-hormonal therapy and chemotherapy [172]; (**C**) biodistribution data of MC7-L1 and MC4-L2 (ER+ and ERαKD, respectively) tumor uptake for FES and 4FM-^18^F-FES [169]. * *p*  <  0.05; ** *p*  <  0.01.
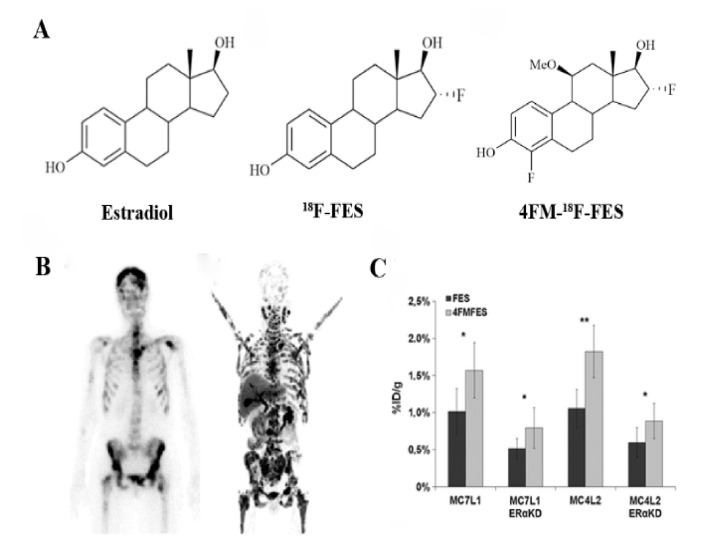


ERβ is an inactive subtype of ER compared with ERα, with relatively fewer studies involving ERβ. The level of ERβ in tumor cells was correlated with breast cancer progression. PET imaging targeting ERβ could also be applied for early diagnosis of breast cancer. There are mainly two types of ERβ probes. One is based on diarylpropionitrile (DPN) pharmacophore with high selectivity and is a structurally modified DPN, labeled with ^18^F and synthesized as ^18^F-FEDPN [173]. PET imaging showed that ^18^F-FEDPN had uptake in the animal uterus and ovary, but its specificity was not strong enough for ERβ imaging. Moon et al. [174] eventually synthesized DPN analogs that contain substituents such as methyl, hydroxymethyl, and fluoroethyl, which had a higher selectivity of ERα/ERβ than DPN. However, the absolute ER-β binding affinity was not sufficient and not available for PET imaging. Therefore, scientists need to develop a more optimized binding affinity and adequate selectivity ligands for in vivo ERβ imaging.

Other types of probes are cyclofenil (C4-^18^F-Fluorocyclofenil, ^18^F-FCF) and its derivatives (C3-^18^F-fluoroethylcyclofenil, ^18^F-FECF). Both types of probes can be prepared with high radiochemical purity, radiochemical yields, and high affinity. However, there was no effective uptake in the target tissue in animal experiments, and the imaging results were unsatisfactory. In 2012, Lee et al. [175] prepared fluorine-18-radiolabeled 8β-(2-fluoroethyl)estradiol (^18^F-8BFEE2) with limited potential for PET imaging of ERβ as well.

In addition to PET probes, many SPECT imaging agents have been successively developed. Marc B. Skaddan et al. [176] first labeled estradiol at the 7α position with ^99m^Tc in 2000. The emergence of the ^99m^Tc(I)-estradiol-pyridin-2-yl hydrazine derivative represents a solid step forward in the design of estrogen-based ^99m^Tc-labeled tracers with improved imaging characteristics [177]. Our group synthesized a novel estradiol-based probe, ^99m^Tc-DTPA-estradiol, with satisfactory labeling efficiency and stability [171]. It showed favorable properties in vitro and in vivo, demonstrating potential use in ER-positive tumor imaging. The in vivo pharmacokinetics of this probe should be optimized further, especially to reduce the liver uptake. Although SPECT imaging of ER is an alternative option for ^18^F-FES PET imaging, it still possesses a low tumor uptake and high background problem. The development of neutral non-steroidal analogs specifically binding to each ER subtype with high-affinity specificity would be a potential development direction.

### 3.2. Targeting Progesterone Receptor (PR)

Approximately 70% of breast cancer patients are estrogen-dependent and half of these patients also have high progesterone expression. Progesterone regulates the function of the reproductive tract as well as target tissues such as the ovary, uterus, and mammary gland [178], while also serving as a precursor for the synthesis of estrogens, androgens, and adrenocortical steroids [179], which play essential hormonal roles in the female reproductive system. Progesterone induces biological effects by combining with its receptors, with PR expression occurring in the brain, pituitary gland, mammary gland, and the female reproductive tract [180]. PR is regulated by an estrogen-related gene, and its expression is highly dependent on ER [181]. Tumors with ER−/PR+ make up less than 1% of all breast cancers [182]. The expression of ER and PR is thought to have a directional effect on the display of functional ER bypass. Studies have shown that nearly 75% of ER+/PR+ tumors respond positively to endocrine therapy, while ER+/PR− tumors are likely to be ineffective [105]. PR status may be a better response variant to endocrine therapy than ER status within metastatic breast cancers [183], meaning breast cancer without PR expression is generally associated with poor prognosis and strong invasiveness [184]. Routinely evaluating PR status is helpful for therapeutic decision making and predicting the prognosis.

In recent years, various radioactive molecular probes targeting PR have been developed. Despite this, only a few radiotracers have been tested in clinical trials, including ^18^F-FENP, ^18^F-FMNP, 6α-^18^F-Fluoroprogesterone, ^18^F-FPTP, and ^18^F-FFNP (Figure 6A). ^18^F-FENP was the first radiofluorinated progestin and possessed a high binding affinity for PR, which was 60 times more potent than that of progesterone [185]. However, its high lipophilicity and metabolic liability led to increased adipose tissue and liver uptake, high background activity, and a low target/background ratio, with metabolic defluorination also resulting in high bone uptake. Thus, ^18^F-FENP is not an ideal PR imaging probe.

^18^F-FMNP is a 16-methyl analog of ^18^F-FENP. Although in competitive binding assays, it displayed high affinity and specificity for PR, ^18^F-FMNP also had high lipophilicity and metabolic liability, and no clinical application has occurred yet [13]. An alternative compound, 6α-^18^F-Fluoroprogesterone, had the same problem, possessing a low uterus uptake because of its high fat uptake and relatively low target tissue selectivity. It also cannot avoid the high bone uptake resulting from metabolic defluorination [186]. As for ^18^F-FPTP, the complicated synthetic process and low radiochemical yield prevent further application [187].

The most promising PR imaging radioligand is ^18^F-FFNP (21-^18^F-fluoro-16α, 17α-[(R)-(1′-α-furylmethylidene)-19-norpregn-4-ene-3, 20-dione), the only PR-based probe that has been clinically evaluated in humans. It can specifically bind to PR with high affinity and high selectivity, and show better imaging effects [188]. In a first-in-human feasibility study, ^18^F-FFNP imaging showed drug safety and organ radiation dose safety among breast cancer patients. The tumor-to-normal breast tissue uptake ratio showed that progesterone-positive tumors had significantly higher uptake than negative tumors. A preclinical study carried out by Fowler et al. [189] revealed an increased uptake of ^18^F-FFNP in SSM3 tumor-bearing mice after using estrogen therapy. This is mainly due to the synergistic stimulation involved in the expression of estrogen-related progesterone genes. Research by Linden et al. [105] also showed that when tamoxifen was used to treat advanced breast cancer patients, there was a synergistic increase at the early stage. The uptake of ^18^F-FFNP was significantly reduced after taking the anti-estrogen drug fulvestrant [189]. Therefore, future studies will focus on anti-estrogen therapy of breast cancer patients, especially patients with reduced estrogen levels, using ^18^F-FFNP PET imaging to evaluate the therapeutic effect. Furthermore, ^18^F-FFNP can be combined with the ER probe ^18^F-FES to improve the diagnostic accuracy further.

In 2017, Wu et al. [14] designed and prepared a novel PR-targeting probe, ^18^F-EAEF (Figure 6A), with a high radiochemical yield, good radiochemical purity, good specificity, and high stability in saline and serum. In biodistribution and PET imaging, PR-expressing tissues within the uterus and ovary had high levels of ^18^F-EAEF accumulation at 2 h post-injection, while muscle uptake was very low. In PR-positive MCF-7 tumors, ^18^F-EAEF had a high uptake, and the tumor-to-muscle ratio was 2.90. Meanwhile, in an EAEF blocking group and a PR-negative MDA-MB-231 control group, tumor uptake was lower (Figure 6C). This shows that ^18^F-EAEF may be a useful PR imaging probe and worth further investigation. However, only limited progress has been made, and more suitable tracers for PR imaging still require further research.

### 3.3. Targeting HER2

Epidermal growth factor receptors (EGFR) play a crucial role in regulating cellular processes, including tumor cell growth and differentiation, proliferation, angiogenesis, and antiapoptotic functions. They can also upregulate the expression of genes that activate epithelial–mesenchymal transition, leading to the initiation of metastasis [190]. HER2 is one member of the EGFR family of tyrosine kinases. About 15–20% of primary breast cancer patients have HER2/erbB2 oncogene overexpression or exhibit amplification, which is associated with aggressive tumor behavior and a poor clinical outcome [191,192]. Therefore, HER2 has become an important prognostic and predictive factor, as well as a target for molecular therapies [193]. Studies have shown that HER2 expression variability is as high as 13% to 30% [194], and patients with different levels of HER2 expression respond differently to tumor therapy [195]. Therefore, it is considered important to monitor HER2 expression levels during HER2-targeted treatment for the classification and efficacy evaluation of breast cancer tumors.

Recently, a series of imaging probes has been used for noninvasive detection and evaluation of breast cancer HER2 expression. ^64^Cu-Trastuzumab, ^64^Cu-DOTA-Z_her2:477_, ^68^Ga-Trastuzumab (Fab’), ^68^Ga-ABY-002, and ^89^Zr-Trastuzumab have all been used for noninvasive detection and evaluation of HER2 expression in breast cancer [193]. Due to its high uptake in the liver and kidney, ^68^Ga-ABY-002, the first reported clinical imaging agent using a non-immunoglobulin-based scaffold protein, was only used for the detection of abdominal tumor metastasis of breast cancer [196]. Compared with ^18^F-FDG, ^68^Ga-DOTA-F(ab’)_2_-trastuzumab PET imaging showed a better downregulation effect of HER2 associated with Hsp90 inhibition [197]. Recently, ^111^In-DPTA-trastuzumab has been used for SPECT imaging to evaluate the efficacy of trastuzumab therapy. However, its low tumor-to-blood ratio limits the specific identification of tumor sites [16]. The positive electron probe ^89^Zr-Trastuzumab displayed a high image quality in preclinical studies and obtained good spatial resolution and sensitivity [15,17]. Due to the long half-life of ^89^Zr, it can be visualized up to 7 days after a single injection. Dijkers et al. [198] first applied ^89^Zr-trastuzumab to the human body, with the PET imaging results clearly showing HER2-positive tumors, as well as liver, lung, bone, and even intracranial metastases (Figure 7). The imaging could still detect occult metastatic lesions 5 days after injecting ^89^Zr-trastuzumab. Perhaps ^89^Zr-trastuzumab will be a potential probe for breast cancer assessment, and further research is ongoing.

### 3.4. Targeting Gastrin-Releasing Peptide Receptor (GRPR)

GRPR, namely, bombesin (BBN) receptor subtype II, is a seven-transmembrane G protein-coupled receptor conjugated with BBN. Over recent decades, GRPR has been found to be highly expressed in various human cancers such as lung, gastric, breast, pancreas, prostate, colorectal, ovarian, and endometrial cancers, and gliomas, but it has a low expression or no expression in normal tissues [199]. It has been reported that GRPR is overexpressed in 65% of ductal carcinomas and 68% of invasive ductal carcinomas of the breast [200]. The amino acid sequence of BBN, Gln-Trp-Ala-Val-Gly-His-Leu-Met-NH2, is identical to the C-terminal of human GRPR. BBN has been extensively investigated in the development of various molecular probes that label different isotopes for the visualization of GRPR expression [201,202]. Preclinical studies have demonstrated these tracers’ potential to evaluate GRPR expression in GRPR-positive tumors [203], with some radiotracers being used in breast cancer imaging [204].

### 3.5. Other Receptors

In addition to the probes mentioned above, other radiolabeled molecular imaging probes have proved effective in breast cancer detection. For example, radionuclide imaging of immune checkpoint proteins helps improve therapeutic efficacy by identifying patients who will potentially benefit from immunotherapy. A preclinical study with ^89^Zr-labeled atezolizumab (an anti-PD-L1 antibody) carried out in a group of 22 patients with bladder cancer, non-small cell lung cancer, and TNBC suggested that the radionuclide probe uptake appeared to be a good predictor of treatment response (Figure 8A) [205]. Somatostatin receptor (SSTR) is also a useful target since SSTR has been reported in breast cancer. Chereau et al. [18] observed that ^68^Ga-DOTA-TOC PET imaging was able to visualize breast cancer xenografts overexpressing SSTR that were barely visible using ^18^F-FDG. Chemokines and their receptors are other target pairs for imaging since a growing body of evidence reveals that their interaction is critical in cancer progression. For example, CXCR4 overexpression has been identified as an adverse prognostic factor in breast cancer. In preclinical studies, tumor uptake of ^64^Cu-AuNCs-AMD3100 correlated with CXCR4 expression in both primary lesions and lung metastases of a mouse 4T1 orthotopic breast cancer model (Figure 8B) [19]. Other targets that are not directly expressed in tumor cells but are highly expressed in tumor vasculatures also exerted moderate potential for breast cancer imaging. Our group synthesized ^99m^Tc-HYNIC-VCAM-1_scFv_ using a single-chain variable fragment (scFv) of anti-vascular cell adhesion molecule-1 (VCAM-1) antibodies with a high radiolabeling yield [20]. In vivo SPECT imaging demonstrated that tumor uptake was observed in a xenograft with human MDA-MB-231 breast cancer. ^99m^Tc-HYNIC-VCAM-1_scFv_ provided a qualitative and semiquantitative method for the noninvasive evaluation of VCAM-1 expression. Cyclin-dependent kinases 4/6 (CDK4/6) control the cell cycle from the G1 to the S phase and are overexpressed in many cancers, including breast cancer. The use of radiolabeled CDK4/6 inhibitor (CDKi) for tumor imaging has gained increased attention. Ramos et al. [206] reported ^18^F-CDKi as a novel PET imaging agent to quantify CDK4/6 expression in ER-positive, HER2-negative breast cancer. Phosphatidylinositol 3-kinase (PI3K) is another intracellular kinase that regulates cell proliferation, survival, and migration, and about 70% of breast cancers have been found to have abnormal activation of PI3K/Akt/mTOR. Our group labeled the PI3K inhibitor GDC-0941 with ^11^C for PET imaging in MCF-7 xenograft models, demonstrating excellent tumor penetration.

### 3.6. Dual Receptor-Targeted Molecular Imaging

For tumor single-target imaging, cell surface receptors must be highly expressed in tumors compared to normal tissues, which may not occur throughout the whole development process of tumors or in all types of breast cancers [21]. Moreover, the relatively weak affinity and pharmacokinetic characteristics in vivo lead to unsatisfactory imaging. Therefore, the development of dual receptor-targeted moleculer imaging agents has gradually drawn researchers’ attention [207]. Compared with monovalent binding, molecular imaging probes based on heterodimers that bind two different biomarkers can significantly improve tumor targeting efficacy [208]. Dual receptor-targeted moleculer imaging agents have greater imaging contrast and higher specificity for diseased tissues due to their increased maximum binding and improved pharmacokinetic profiles.

Recently, Liu et al. designed a heterodimeric peptide that specifically targeted integrin αvβ3 and GRPR–Glu-c(RGDyK)-bombesin (RGD-BBN) [209]. This single molecule contains two promising pharmaceuticals that can be used to sensitively detect tumors as long as a high expression of integrin αvβ3 or GRPR is present. RGD-BBN was labeled with three different radionuclides (^18^F, ^64^Cu, and ^68^Ga), and the tumor targeting affinity and pharmacokinetics of their corresponding PET radiotracers in breast cancer models were studied. Micro-animal PET imaging results demonstrated that all three radiotracers possessed both integrin αvβ3 and GRPR binding affinity in vitro and can visualize the tumor areas. Compared to the other two probes, ^18^F-FB-PEG3-RGD-BBN had the lowest tumor uptake rate, and its synthesis method was the most complex. Although ^68^Ga-NOTA-RGD-BBN (Figure 9) showed high tumor signals, its background uptake was also relatively high. Zhang et al. [210] reported that ^68^Ga-NOTA-RGD-BBN could discern primary and metastatic breast cancers and may potentially be used in breast cancer diagnosis, staging, and surgery guidance. ^64^Cu-NOTA-RGD-BBN had a prolonged tumor uptake time, but also higher liver retention. Despite this, it exhibited significantly higher tumor uptake and improved in vivo kinetics compared to its corresponding RGD and BBN monomers or the combined mixture [211]. As mentioned earlier, this is precisely due to the advantages of dual-target imaging. Even if only one receptor is expressed, ^64^Cu-NOTA-RGD-BBN can create a good image.

For SPECT, Liu et al. [22] synthesized ^99m^Tc-RGD-BBN, and its ability to identify tumors (2.69 ± 0.66% ID/g, 1 h post-injection) from inflammation (1.20 ± 0.32% ID/g) was superior to ^18^F-FDG. Furthermore, ^99m^Tc-3P4-RGD2 can only detect GRPR-positive cancer, while ^99m^Tc-RGD-BBN can successfully detect tumors when integrin αvβ3 or GRPR is not expressed at the same time [21]. In addition, Chen et al. [21] studied the safety, radiation dosimetry, and diagnostic performance of ^99m^Tc-RGD-BBN with six female breast cancer patients and six healthy volunteers for the first time. This probe exhibited good stability and excellent properties for detecting breast cancer. This preliminary study demonstrated the powerful potential of ^99m^Tc-RGD-BBN in unknown breast cancer diagnosis.Figure 9(**A**) chemical structure of ^68^Ga-BBN-RGD. (**B**) ^68^Ga-BBN-RGD PET/CT in patient with invasive ductal carcinoma (**a**–**c**). The lesion had a strong GRPR expression (**d**) and weak integrin αvβ3 expression (**e**). (**C**) A 57-year-old woman who underwent breast cancer radical mastectomy and follow-up with ^68^Ga-BBN (**a**) and ^68^Ga-BBN-RGD (**b**) PET/CT, respectively. ^68^Ga-BBN-RGD PET/CT had a significantly higher SUVmax in tumor lesions than that of ^68^Ga-BBN PET/CT [210].
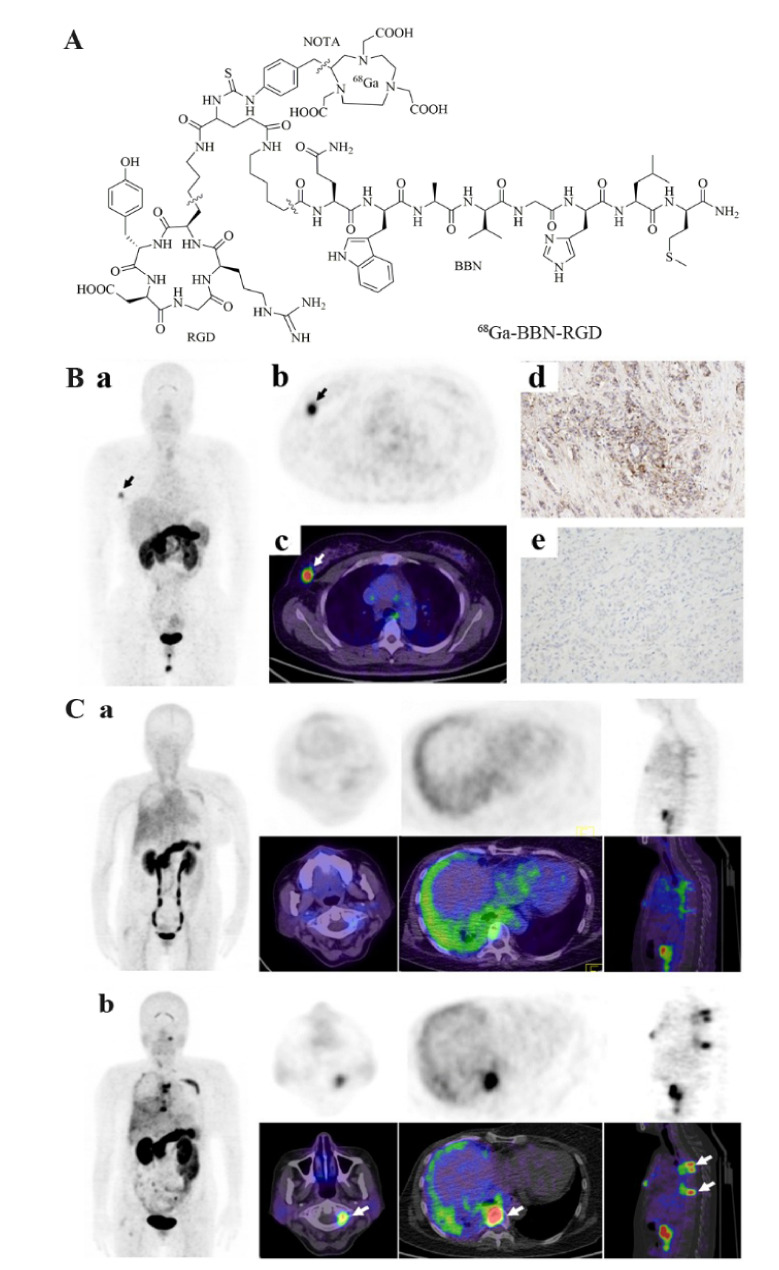


Our group developed a heterodimeric tracer consisting of RGD and asparagine−glycine−arginine (NGR) peptides for PET imaging of breast cancer targeting αvβ3 and CD13, respectively. Compared with monomeric ^68^Ga-NGR and ^68^Ga-RGD, the dual receptor-targeting tracer ^68^Ga-NGR-RGD showed higher binding affinities and targeting efficiency, and a longer tumor retention time [212]. Additionally, amivantamab is a novel bispecific antibody that simultaneously targets EGFR and the hepatocyte growth factor receptor (HGFR/c-MET) that are overexpressed in TNBC. In a recent report, Cavaliere et al. [213] radiolabeled amivantamab with ^89^Zr, and it demonstrated higher tumor uptake than that of the radiolabeled single-arm parent antibodies.

## 4. Biomaterial-Based Probes for Imaging of Breast Cancer

Biomaterial-based probes include membrane-based, exosome-based, and peptide nucleic acid-based imaging probes.

### 4.1. Membrane-Based Imaging Probes

The development of nanotechnology prompted its application in tumor diagnosis and therapy [23]. However, synthetic nanoparticles are often rapidly recognized and eliminated by the mononuclear phagocyte system when exposed to body fluids [214]. Therefore, modifying nanoparticles is particularly important for reducing their uptake within the reticuloendothelial system. For this purpose, surface functionalization of nanoparticles with polymer materials [215] and cell membranes [216] represents two representative approaches. However, the effectiveness and safety of synthetic polymers remain problematic due to blood clotting, cell agglutination, and the production of PEG antibodies [217]. In contrast, using cell membranes as a natural, safer, and more biocompatible vector offers lower immunogenicity. Recently, the application of cell membranes for nanoparticle surface functionalization has become a focus of research, such as the erythrocyte membrane [218], platelet membrane [219], and cancer cell membrane [220].

Our group used the cancer cell membrane of MDA-MB-231 to modify upconversion nanoparticles (CCm_231_-UCNPs) for in vivo multimodality imaging of TNBC [221]. This probe exhibited homologous targeting and immune-escaping abilities, and performed well in breast cancer molecular classification which can successfully differentiate MDA-MB-231 in MCF-7 tumor-bearing mouse models in vivo (Figure 10A). The probe has the potential to be used as a drug delivery system for the treatment of TNBC. In addition, our group designed RBC-derived membrane-coated UCNPs, which can be used for the targeted UCL/MRI/PET tri-modality imaging of 4T1 breast cancer [24]. The ability of tumor targeting was obtained by inserting FA into the cell membranes (Figure 10B). Our team creatively realized the efficacy of 4T1 tumor PET imaging based on a short half-life radionuclide-labeled biomimetic nanoparticle, using pre-targeting strategies and an in vivo click chemistry methodology. Our work provides a new direction for multiple clinical applications of these biomimetic nanoparticles. The biomimetic cell membrane-coated nanoparticles have the potential for future imaging and treatment of diseases. However, there are also many limitations and challenges that need to be elucidated in more detail, including long-term safety and stability and the sourcing and purification efficiency of the cell membrane.

### 4.2. Exosome-Based Imaging Probes

Over the years, exosomes have attracted tremendous attention for carrying cellular proteins and genetic information and facilitating antigen presentation [25]. Exosomes are small, naturally secreted membrane vesicles consisting of a lipid bilayer, with a size of 40–100 nm [222]. Many different cells such as immune cells, epithelial cells, mesenchymal cells, and tumor cells can release exosomes, and researchers have successfully isolated them from blood plasma, serum, and urine. Exosomes are involved in intercellular communication, immune response, and cancer metastasis and have been used as a promising natural drug delivery vector due to their biocompatibility and inherent targeting ability [223].

The radiolabeling of exosome vesicles for SPECT imaging, with the purpose of studying their biodistribution in vivo imaging performance, has only been reported in a few studies [224,225,226]. Shi et al. [227] first used ^64^Cu-radiolabeled PEG-modified exosomes and studied their systemic biodistribution. Compared with traditionally reported native exosomes, ^64^Cu−NOTA−exosome−PEG reduced premature hepatic clearance, prolonged blood circulation, and exhibited higher accumulation in a 4T1 tumor after 24 h post-injection. Therefore, the radiolabeling of exosomes is a reliable and accurate approach to tumor imaging, with exosomes showing promising potential as novel theranostic vesicles.

### 4.3. Peptide Nucleic Acid-Based Imaging Probes

It is well known that the occurrence and progression of cancer are regulated by genes. Peptide nucleic acids (PNAs) are antisense oligonucleotides that hybridize more strongly and specifically to RNA and DNA, resisting attack by both nucleases and proteases [26]. PNAs do not induce RNase degradation of bound RNA but solely inhibit mRNA translation, which provides an opportunity for molecular imaging and gene therapy. Noninvasive antisense imaging with high sensitivity and specificity using PNAs could image oncogene expression in vivo and further determine cell malignancy at an early stage.

Tian et al. [228] showed scintigraphic detection of CCND1 mRNA with a ^99m^Tc-chelator-PNA-peptide probe in MCF7 breast cancer xenografts in mice. Paudyal et al. [27] synthesized a ^64^Cu-DOTA-PNA-peptide targeting HER2 mRNA expression. They determined that treatment effectiveness or resistance in human BT474 xenografts could be detected sooner than the currently available modalities, such as CT or MRI. Our group successfully prepared a ^99m^Tc-labeled PNA sequence that undergoes complementary binding to the oncogene miR-155 [229]. Compared to [^99m^Tc]mis-PNA, [^99m^Tc]anti-PNA-155 can visualize MCF-7 tumors with a relatively high expression of miR-155. When injected with excessive anti-PNA-155, the tumor visibility of MCF-7 was less visible, suggesting better probe specificity. This provides helpful information at the gene level for breast cancer imaging, but the relatively higher radioactivity in the blood and relatively low tumor uptake still require further improvement.

## 5. Conclusions and Future Perspectives

Molecular imaging is a rapidly emerging technology that allows for noninvasive imaging of receptor expression at the molecular level and creates the possibility to diagnose breast cancer accurately. Since breast cancer is a heterogeneous tumor and the expression of markers can vary as the disease progresses, radiotracer selection according to imaging purposes appears to be particularly important. As mentioned in the article, various molecular probes have their advantages and disadvantages. As novel markers are identified and new imaging probes are continuously developed, molecular imaging can become an indispensable tool in oncology.

## Figures and Tables

**Figure 1 cancers-13-05459-f001:**
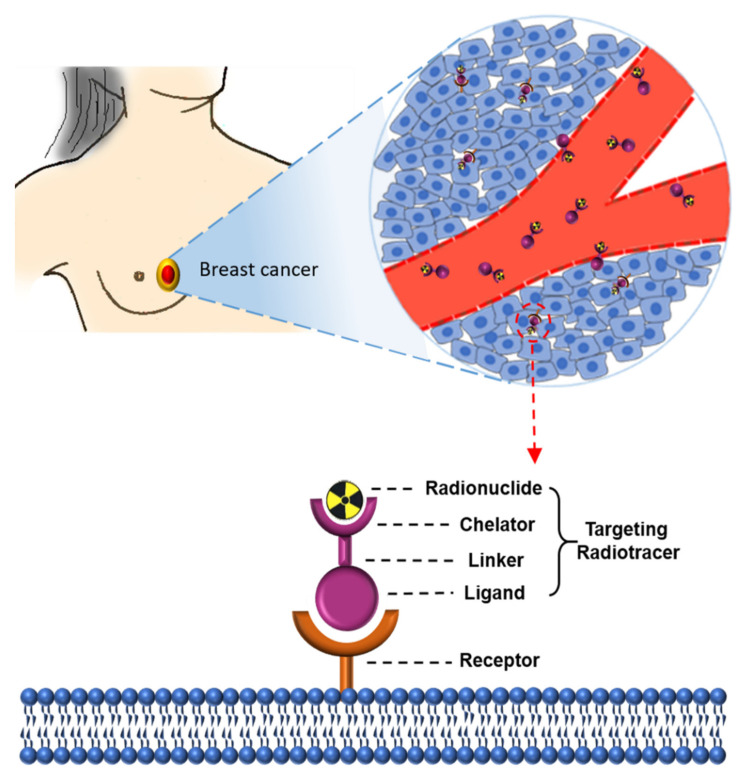
Schematic overview of receptor-targeting molecular imaging for breast cancer. The molecular probes consist of a ligand, linker, chelator, and radionuclide. Ligands that bind to the overexpressed receptors on breast cancer cells can be coupled to a chelator often through a linker. Chelators enable the labeling of ligands with radionuclides such as ^68^Ga and ^99m^Tc, which are combined through a linker.

**Figure 2 cancers-13-05459-f002:**
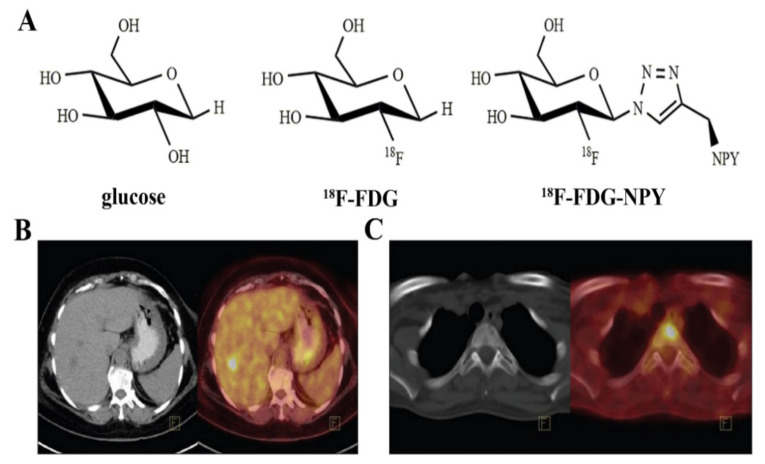
Chemical structure of (**A**) glucose, ^18^F-FDG, and ^18^F-FDG-NPY; (**B**) an invasive ductal carcinoma patient with a liver lesion at the right lobe, segment 5 [33]; (**C**) a female patient with infiltrating lobular carcinoma. ^18^F-FDG PET/CT detected multiple bone metastases [33].

**Figure 3 cancers-13-05459-f003:**
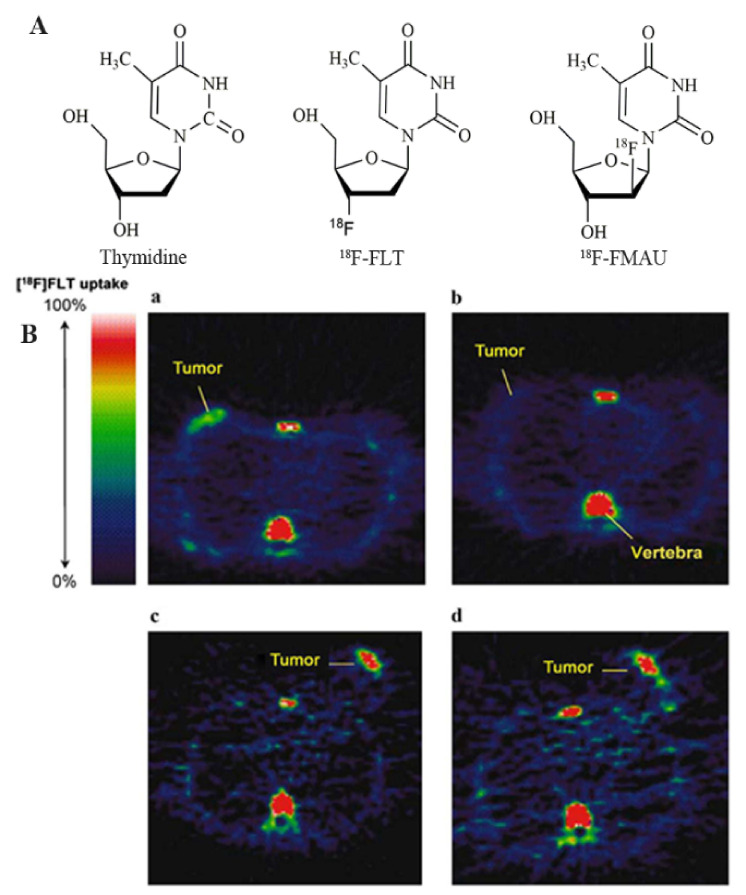
Chemical structure of (**A**) thymidine, ^18^F-FLT, and ^18^F-FMAU; (**B**) ^18^F-FLT PET imaging of a patient with grade II breast cancer 1 week after administration of combination chemotherapy. (**a**) pretreatment and (**b**) posttreatment of a patient with lobular breast cancer who responded to treatment. (**c**) pretreatment and (**d**) posttreatment of a patient with invasive ductal breast cancer who did not respond to treatment [104].

**Figure 4 cancers-13-05459-f004:**
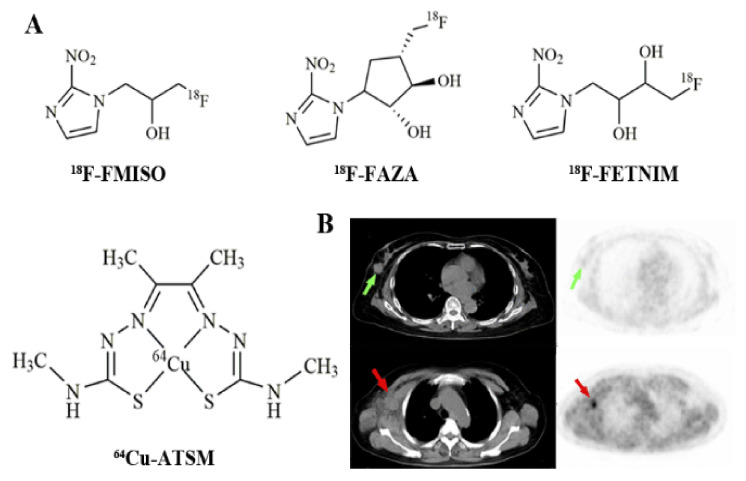
Chemical structure of (**A**) ^18^F-FMISO, ^18^F-FAZA, ^18^F-FETNIM, and ^64^Cu-ATSM; (**B**) ^18^F-FMISO PET imaging of a patient with right breast tumor in situ (green arrow), CT and PET images, and a patient with right axillary node metastasis (red arrow), CT and PET images [114].

**Figure 6 cancers-13-05459-f006:**
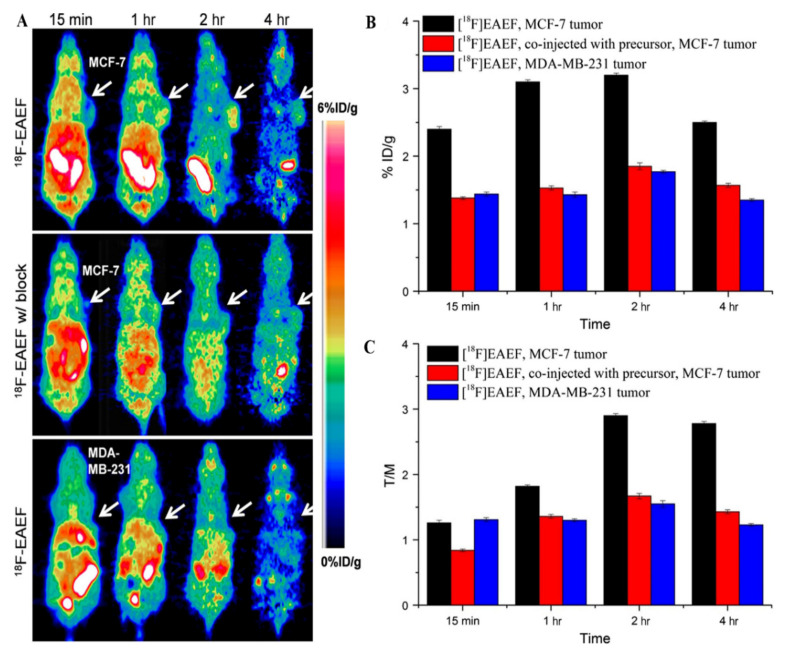
Chemical structure of PET imaging and tumor uptakes in PR-positive MCF-7 and PR-negative MDA-MB-231 tumor-bearing mice at different times after injection of ^18^F-EAEF. (**A**) micro-PET images of MCF-7 tumor with ^18^F-EAEF (up), co-injected excessive precursor (middle), and MDA-MB-231 tumor with ^18^F-EAEF (bottom); (**B**) tumor uptakes and (**C**) tumor-to-muscle ratios at 15 min, 1 h, 2 h, and 4 h post-injection according to PET imaging [14].

**Figure 7 cancers-13-05459-f007:**
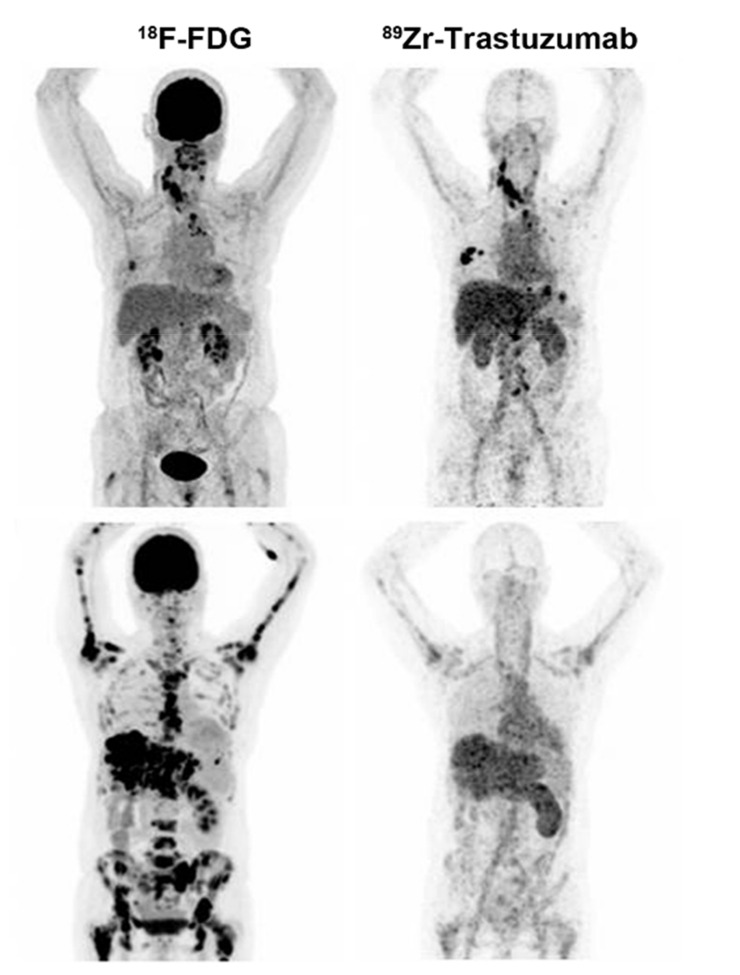
^18^F-FDG and ^89^Zr-trastuzumab PET scans of a patient with a ^89^Zr-trastuzumab PET scan considered HER2-positive breast cancer (**up**) and a ^89^Zr-trastuzumab PET scan considered HER2 negative (**bottom**) [15].

**Figure 8 cancers-13-05459-f008:**
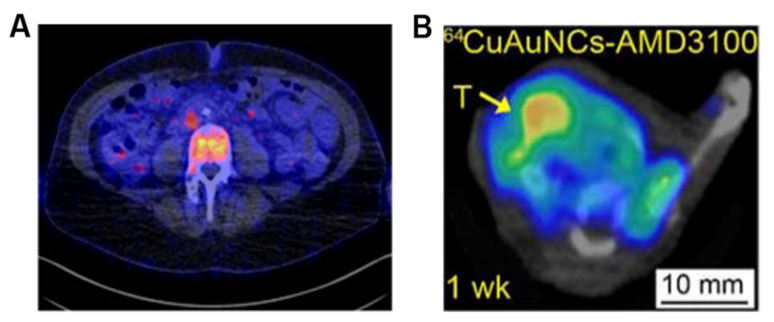
Other radiolabeled molecular imaging probes to detect breast cancer. (**A**) PET/CT image of a bone metastasis in a TNBC patient with heterogeneous intralesional ^89^Zr-atezolizumab uptake on day 7 post-injection [205]. (**B**) representative PET/CT image of ^64^Cu-AuNCs-AMD3100 showing the tracer accumulation in a 4T1 tumor model at 1 week post-tumor implant [19].

**Figure 10 cancers-13-05459-f010:**
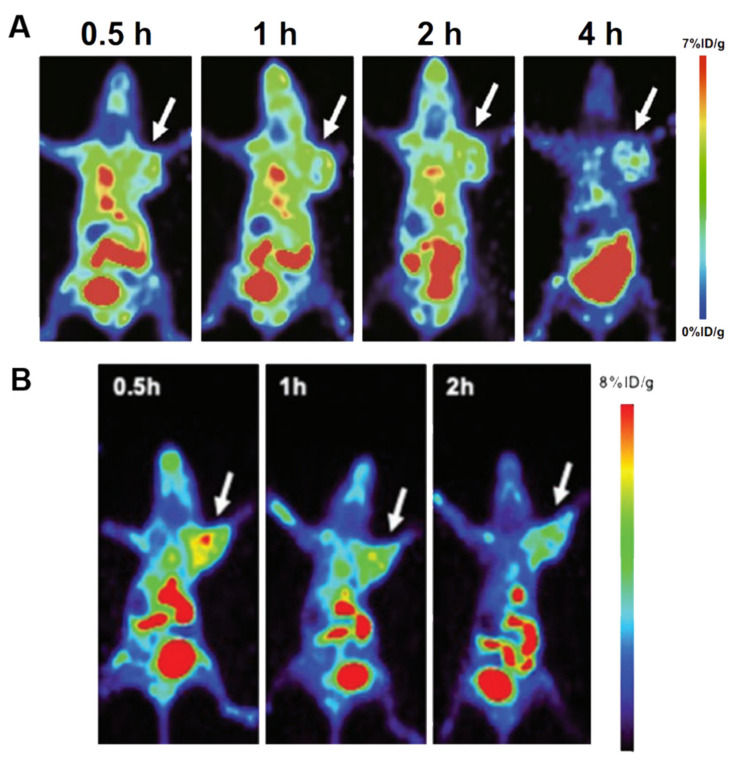
(**A**) micro-PET static imaging was performed at 0.5, 1, 2, and 4 h after injection of Al^18^F-L-NETA-DBCO in MDA-MB-231 tumor-bearing mice pretreated with N3-DSPE-PEG-CCm231-UCNPs [221]. (**B**) PET imaging of 4T1 breast cancer-bearing mice at different time points in the N3(FA)-PEG-DSPE-RBC-UCNP group after injection of DBCO-L-NETA-Al^18^F [24].

**Table 1 cancers-13-05459-t001:** Summary of various radiotracers for breast cancer imaging.

Receptor Type	Target	Imaging Modality	Imaging Targeting Agents	Examples	Characteristics	Limitations	Ref.
Imaging biological processes	Glycolysis	PET	Glucose analog	^18^F-FDG	1. Reflects cellular glycolysis 2. Has been applied in breast cancer screening, staging, molecular subtype determination, and treatment monitoring	1. Not supposed to diagnose inflammatory breast cancer 2. Limited spatial resolution	[6]
	Amino acid transporter	PET	Methionine	^11^C-MET	Can be used for predicting the early treatment response	Short half-life of ^11^C	[7]
			Leucine analog	^18^F-fluciclovine	1. Long half-life 2. Detects bone, lung, brain, and axillary nodal metastases	Limitations in detecting liver metastases	[8]
	Cellular proliferation	PET	Thymidine analog	^18^F-FLT	Can visualize the status of cell proliferation and avoid the false positive results occurring in ^18^F-FDG imaging	Physiological uptake occurs in highly proliferative tissues	[9]
	Hypoxia	PET	Small molecules	^18^F-FMISO	Evaluation of tumor hypoxia in vivo	1. Slow clearance from the blood 2. Modest hypoxic-to-normoxic ratio and limited contrast images	[10]
Imaging receptors	ER	PET	Estradiol	^18^F-FES	The sensitivity and specificity of ^18^F-FES for tumor detection were 69–100% and 80–100%	1. Lack of precise SUV (standardized uptake value) thresholds to distinguish specific uptake from nonspecific uptake 2. Less selectivity for ERα and ERβ	[11]
		SPECT	Estradiol	^99m^Tc-DTPA-estradiol	Satisfactory labeling efficiency and stability	High background/liver uptake	[12]
	PR	PET	Progestin	^18^F-FENP	A high binding affinity for PR	1. High lipophilicity and metabolic liability led to increased adipose tissue and liver uptake 2. Low target/background ratio	[13]
			Progestin	^18^F-FFNP	Specifically binds to PR with high affinity and high selectivity	Small sample size	[14]
	HER2	PET	Antibody	^89^Zr-trastuzumab	1. Radiolabeling efficiency: 77.6 ± 3.9% 2. Radiochemical purity: 98.1 ± 1.1%	1. Low sensitivity 2. Liver and spleen had higher uptake	[15]
			Antibody fragments	^68^Ga-DOTA-F(ab’)_2_-trastuzumab	Can identify HER2 downregulation by Hsp90 inhibition	Lack of sufficient sensitivity for clinical use	[16]
		SPECT	Antibody	^111^In-DPTA-trastuzumab	1. High stability 2. High labeling yields 3. Maintains immunoreactivity and internalization properties	1. Low sensitivity 2. Low tumor-to-blood ratio 3. Liver, kidney, and spleen had tracer uptake	[17]
	Other receptors	PET	Antibody	^89^Zr-labeled atezolizumab (targeting PD-L1)	Can help assess the PD-L1 status and clinical response predictions	1. A small patient population 2. No tumor biopsies that were immunohistochemically highly PD-L1 positive	[18]
		PET	Peptide	^68^Ga-DOTA-TOC (targeting SSTR)	Could be used for the detection of breast tumors not detected with ^18^F-FDG		[19]
		PET	CXCR4 antagonist	^64^Cu-AuNCs-AMD3100 (targeting CXCR4)	1. Flexible and straightforward preparation 2. High radiolabeling specific activity 3. Sensitive and accurate detection of CXCR4	The ability to determine tumor progression and burden needs further improvement	[20]
		SPECT	scFv	^99m^Tc-HYNIC-VCAM-1_scFv_ (targeting VCAM-1)	The probe can reach the tumor site quickly based on the high tissue penetrability of small antibody fragments	High activity in blood and liver	[21]
	Dual receptor targeted	PET	Peptide	^64^Cu-NOTA-RGD-BBN (targeting αvβ3 and GRPR)	Favorable in vivo kinetics and enhanced tumor uptake	Did not set other controls such as the RAD-bombesin heterodimer and RGD-scramble bombesin heterodimer	[22]
		PET	Peptide	^68^Ga-NGR-RGD (targeting αvβ3 and CD13)	Dual receptor-targeting tracers showed higher binding avidities, targeting efficiency, and longer tumor retention time	The uptake of ^68^Ga-NGR-RGD in tumors is still relatively low	[23]
Biomaterial-based probes	Membrane	PET	Cancer cell membrane	CCm-UCNPs	1. Exhibited homologous targeting and immune escaping abilities 2. Can be used for ultra-sensitive in vivo UCL/MRI/PET multimodality precise imaging of triple-negative breast cancer (TNBC)		[24]
		PET	Red blood cell membrane	RBC-UCNPs	1. The combination of a pre-targeting strategy and in vivo click chemistry successfully realized 4T1 tumor PET imaging by short half-life nuclide-labeled biomimetic nanoparticles 2. The inserted FA was used to increase the tumor-targeting ability of RBC-UCNPs		[25]
	Exosomes	PET	Exosome	^64^Cu-NOTA-exosome-PEG	1. One of the first examples of radiolabeling and in vivo PET imaging of exosomes 2. PEGylation reduced hepatic clearance of exosomes 3. Exhibited enhanced tumor uptake and imaging capacity	The radiolabeling yield of NOTA−exosome−PEG was slightly lower than that of NOTA−exosome	[26]
	Peptide nucleic acid	SPECT	PNA	^99m^Tc-CCND1 antisense probe	Establish the proof of principle for identifying oncogene activity in breast cancer xenografts		[27]

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
