# Peer review of "Radionuclide-Based Imaging of Breast Cancer: State of the Art"

_cancers, 2021, doi:10.3390/cancers13215459_

Round 1

Reviewer 1 Report

This is an excellent manuscript which should be accepted following review be an English speaking native.

Author Response

We sincerely thank the reviewer for your careful reading and positive feedback on our work. The manuscript has been reviewed and edited by a native English speaker. We have accepted the modification suggestions from the professional English speaker in this revision, and we believe that the quality of this manuscript has been improved a lot.

Reviewer 2 Report

The review addresses a wide issue and tries to focus on the role of nuclear medicine in patients with breast cancer. However, authors analyse innovative and promising PET procedures for research purpose but of limited value in clinical practice. 

The introduction, as well as table 1 , appears redundant according to this perspective.  In particular the role of traditional diagnostic imaging is explained by using some general assumptions that don’t properly represent the clinical approach to patients with BC. 

The section on glucose metabolism includes a lot of potential indications including BC screening in healthy women: all these aspects should be critically analysed, in order to evaluate the radiation protection and clinical utility indicating those with a stated role in the diagnostic algorithm for BC. This section, as well as the overall article, should be rewritten excluding clinical PET applications; on the other hand, the sections on innovative and experimental PET tracers are very interesting and should be emphasized. 

Author Response

Response to Reviewer 2

Comment #1: The review addresses a wide issue and tries to focus on the role of nuclear medicine in patients with breast cancer. However, authors analyse innovative and promising PET procedures for research purpose but of limited value in clinical practice.

  1. The introduction, as well as table 1, appears redundant according to this perspective. In particular, the role of traditional diagnostic imaging is explained by using some general assumptions that don’t properly represent the clinical approach to patients with BC.

Response: We sincerely thank the reviewer for the insightful suggestion. To concentrate on the topic of this review, we have deleted the redundant content within the introduction and deleted the old table 1. The modified the corresponding content in our revised manuscript with colored text (red). For your convenience, we also copied the revised paragraph here.

“Breast cancer is one of the most commonly diagnosed malignant tumors, possessing high incidence and mortality rates that threaten women's health [1]. It is the principal cause of cancer-related deaths among females in 2018 [2]. Thus, early and effective breast cancer diagnosis is crucial for enhancing the survival rate. Traditional diagnostic imaging, such as mammography, ultrasound, magnetic resonance imaging (MRI), computed tomography (CT), is based on changes in the anatomical structure of tumors called morphometric imaging. However, breast cancer is a heterogeneous carcinoma. Different types possess different biological features that ultimately affect prognosis, therapy response, and relapse rates. Their invasion and metastasis are closely related to tumor cells and the variety of biomarkers found in the tumor micro-environment. Histological analysis is the primary method used to determine the expression of molecular markers. However, it is limited by sampling a single site at a single time point, which does not sufficiently address tumor heterogeneity [3]. Besides, this progress is invasive and may carry a series of surgical complications including seroma, axillary lymphedema, and local wound infection. As a result, noninvasive molecular imaging studies have been rapidly developed in order to obtain more comprehensive biological tumor information and detect lesions earlier [4].”

Comment #2: The section on glucose metabolism includes a lot of potential indications including BC screening in healthy women: all these aspects should be critically analysed, in order to evaluate the radiation protection and clinical utility indicating those with a stated role in the diagnostic algorithm for BC. This section, as well as the overall article, should be rewritten excluding clinical PET applications; on the other hand, the sections on innovative and experimental PET tracers are very interesting and should be emphasized.

Response: Thank you for your insightful comments. We have revised it accordingly in the section on glucose metabolism and emphasized the sections on innovative and experimental PET tracers in our revised manuscript with colored text (red). The following revised paragraphs are presented here for your convenience. 

18F-FDG imaging can be used for breast cancer diagnosis, staging, molecular subtype, and treatment monitoring. The European Society for Medical Oncology guidelines [34] recommend using 18F-FDG in early-stage breast cancer when conventional examination methods are inconclusive. A study held in Japan showed that the sensitivity and positive predictive value (PPV) of 18F-FDG PET screening for breast cancer were 83.9% and 41.7%, respectively [35]. However, for asymptomatic patients with early-stage disease, the National Comprehensive Cancer Network has not recommended systemic imaging [36]. Additionally, Bertagna et al. [37] indicated the SUV alone should not be used to differentiate between malignant and benign incidentalomas because there is overlap between their SUVs, and 18F-FDG PET/CT is not routinely used for diagnosis of primary breast cancer [38]. For staging, Han et al. [39] performed a systematic review and meta-analysis to evaluate the impact of 18F-FDG on staging and management as an initial staging modality of breast cancer. The results suggested that routine clinical use of 18F-FDG PET, PET/CT, or PET/MRI imaging leads to significant modification of initial staging in newly diagnosed breast cancer patients. A meta-analysis revealed 18F-FDG PET/MRI demonstrates high diagnostic value in the TNM staging in breast cancer patients and can serve as a promising imaging biomarker for future evaluation of the TNM stage of breast cancer [40].”

“For treatment monitoring, 18F-FDG has been widely applied in the management of neoadjuvant chemotherapy (NAC) for locally advanced breast cancer patients. It can effectively monitor therapeutic response and improve the quality of patients’ life [45]. A meta-analysis in 2013 concluded that 18F-FDG imaging can accurately predict NAC’s curative effect on breast cancer in the early-to-mid stage, which has moderately high sensitivity and specificity [46]. Han et al. [47] also reported 18F-FDG provided significant predictive value for the evaluation of responses to NAC in breast cancer patients and might guide rational management. Caldarella et al. [48] performed a meta-analysis of 8 studies with 873 suspected breast cancer cases and came to a similar conclusion. In an international multi-center prospective study, Gebhart et al. [49] evaluated the efficacy of lapatinib and trastuzumab on developing breast cancer patients. 18F-FDG imaging was performed respectively in the baseline period before treatment, the second and sixth weeks after treatment. The results showed a correlation between 18F-FDG uptake at 2 weeks and 6 weeks after treatment, meaning patients who are effective in the second week after targeted therapy are usually effective in the sixth week after treatment. This study indicates that 18F-FDG PET imaging can predict the efficacy of targeted therapy at the early stage, without waiting for the middle stage or the end of treatment. Therefore, 18F-FDG PET imaging has been included in future studies as an essential biological detection method, providing a reference for the clinical decision of NAC and endocrine therapy. However, the specific threshold value [50] and the definition of good histopathologic response varies [51], and the optimal timing of interim PET is unclear [52]. These disparities need to be standardized.”

“Cyclin-dependent kinases 4/6 (CDK4/6) controls the cell cycle from G1 to S phase and is overexpressed in many cancers, including breast cancer. The use of radiolabeled CDK4/6 inhibitor (CDKi) for tumor imaging has gained increased attention. Ramos et al. [183] reported 18F-CDKi as a novel PET imaging agent to quantify CDK4/6 expression in ER-positive, HER2-negative breast cancer. Phosphatidylinositol 3-kinase (PI3K) is another intracellular kinase that regulates cell proliferation, survival, and migration and about 70% of breast cancers has been found to have abnormal activation of PI3K/Akt/mTOR. Our group labeled the PI3K inhibitor GDC-0941 with 11C for PET imaging in MCF-7 xenograft models, demonstrating excellent tumor penetration.”

“Our group developed a heterodimeric tracer consisting of RGD and aspara-gine−glycine−arginine (NGR) peptides for PET imaging of breast cancer targeting αvβ3 and CD13, respectively. Compared with monomeric 68Ga-NGR and 68Ga-RGD, dual-receptor targeting tracer 68Ga-NGR-RGD showed higher binding avidities, targeting ef-ficiency, and longer tumor retention time [189]. Additionally, amivantamab is a novel bispecific antibody that simultaneously targets EGFR and the hepatocyte growth factor receptor (HGFR/c-MET) that are overexpressed in TNBC. In a recent report, Cavaliere et al. [190] radiolabeled amivantamab with 89Zr, and it demonstrated higher tumor up-take than those of the radiolabeled single-arm parent antibodies.”

Reviewer 3 Report

Authors reviewed various methods of radionuclide molecular imaging for the early breast cancer diagnosis. This review paper is well-written and clearly organized with nice summary tables and figures. A couple of comments are listed below.

Comments:

Authors should discuss and review the commonly-used statistical methods in breast imaging. For example, the multiplicity correction methods in breast
MRI, spatial analysis, voxel-wise and cluster-wise methods, univariate and multivariate analysis, etc.

Authors should discuss the popularity comparions among different breast imaging methods. It will be nice if authors can collect some data based on meta analysis to draw the conclusion about the method popularity.

Author Response

Response to Reviewer 3:

Authors reviewed various methods of radionuclide molecular imaging for the early breast cancer diagnosis. This review paper is well-written and clearly organized with nice summary tables and figures. A couple of comments are listed below.

Comment #1: Authors should discuss and review the commonly-used statistical methods in breast imaging. For example, the multiplicity correction methods in breast MRI, spatial analysis, voxel-wise and cluster-wise methods, univariate and multivariate analysis, etc.

Response: We thank the reviewer for the very helpful suggestions for improving our manuscript. According to your suggestion, the commonly-used statistical methods in breast imaging were discussed.

Protocol optimization and appropriate statistical methods and image post-processing techniques will help improve the clinical utility and scientific research. For example, the ability to replicate the results of imaging is essential to the progress of biomedical research. The reproducibility of MRI is affected by multiple comparison correction strategies. Chen et al. [1] reported how to select multiple comparison correction strategies to enhance reproducibility in MRI. Pediconi et al. [2] performed the correction for multiplicity according to the Bonferroni criterion when retrospective analyzed the value of MRI in comparison to X-ray mammography and ultrasound in suspicious breast cancer patients with dense breast parenchyma. Whisenant et al. [3] performed a secondary analysis of the A6702 multicenter trial and assessed the impact of image quality on the diagnostic performance of breast diffusion-weighted imaging (DWI). Multiplicity was also controlled using a Bonferroni correction. Additionally, to evaluate any association of tumor apparent diffusion coefficient (ADC) values with axillary lymph node metastasis (ALNM) in early-stage invasive ductal carcinoma, Kim et al. [4] reviewed retrospectively 270 invasive ductal carcinoma patients who underwent MRI and used multivariate regression analysis and receiver operating characteristic (ROC) curve analysis to analyze the value of tumor apparent diffusion coefficient (ADC) for predicting ALNM. The results showed lower tumor ADC values are associated with the presence of ALNM in early-stage invasive ductal carcinoma.

Comment #2: Authors should discuss the popularity comparisons among different breast imaging methods. It will be nice if authors can collect some data based on meta analysis to draw the conclusion about the method popularity.

Response: We greatly appreciate your constructive comments that help us to improve our review comprehensively. We have discussed the popularity comparisons among different breast imaging methods accordingly, and revised our revised manuscript with colored text (red). For your convenience, please also refer to the following revised paragraphs here.

Zhang et al. [5] reported a network meta-analysis (NMA) to determine the diagnostic accuracy of different imaging technologies (mammography, ultrasound, and MRI) for breast cancer and to help clinicians make more accurate diagnosis decisions. The results will be published in a peer-reviewed journal and we will keep an eye on its results. Kim et al. [6] performed a NMA of 22 direct comparison studies to compare diagnostic performances of 8 different imaging modalities for preoperative detection of axillary lymph node metastasis of breast cancer. The results showed that elastography had the best sensitivity, specificity, positive predictive value, negative predictive value, accuracy, and diagnostic odds ratio. Pediconi et al. [2] retrospective analyzed the value of MRI in comparison to x-ray mammography and ultrasound for breast cancer evaluation in patients with dense breast parenchyma and confirmed breast MRI is significantly superior to mammography and ultrasound. Zhang et al. [7] assessed the diagnostic accuracy of PET/CT and MRI for ALNM in the same population of breast cancer by a systematic review and meta-analysis including eleven studies. PET/CT and MRI had a comparable diagnostic performance with low sensitivity and high specificity for the detection of ALNM.

18F-FDG imaging can be used for breast cancer diagnosis, staging, molecular subtype, and treatment monitoring. The European Society for Medical Oncology guidelines [34] recommend using 18F-FDG in early-stage breast cancer when conventional examination methods are inconclusive. A study held in Japan showed that the sensitivity and positive predictive value (PPV) of 18F-FDG PET screening for breast cancer were 83.9% and 41.7%, respectively [35]. However, for asymptomatic patients with early-stage disease, the National Comprehensive Cancer Network has not recommended systemic imaging [36]. Additionally, Bertagna et al. [37] indicated the SUV alone should not be used to differentiate between malignant and benign incidentalomas because there is overlap between their SUVs, and 18F-FDG PET/CT is not routinely used for diagnosis of primary breast cancer [38]. For staging, Han et al. [39] performed a systematic review and meta-analysis to evaluate the impact of 18F-FDG on staging and management as an initial staging modality of breast cancer. The results suggested that routine clinical use of 18F-FDG PET, PET/CT, or PET/MRI imaging leads to significant modification of initial staging in newly diagnosed breast cancer patients. A meta-analysis revealed 18F-FDG PET/MRI demonstrates high diagnostic value in the TNM staging in breast cancer patients and can serve as a promising imaging biomarker for future evaluation of the TNM stage of breast cancer [40].”

“For treatment monitoring, 18F-FDG has been widely applied in the management of neoadjuvant chemotherapy (NAC) for locally advanced breast cancer patients. It can effectively monitor therapeutic response and improve the quality of patients’ life [45]. A meta-analysis in 2013 concluded that 18F-FDG imaging can accurately predict NAC’s curative effect on breast cancer in the early-to-mid stage, which has moderately high sensitivity and specificity [46]. Han et al. [47] also reported 18F-FDG provided significant predictive value for the evaluation of responses to NAC in breast cancer patients and might guide rational management. Caldarella et al. [48] performed a meta-analysis of 8 studies with 873 suspected breast cancer cases and came to a similar conclusion. In an international multi-center prospective study, Gebhart et al. [49] evaluated the efficacy of lapatinib and trastuzumab on developing breast cancer patients. 18F-FDG imaging was performed respectively in the baseline period before treatment, the second and sixth weeks after treatment. The results showed a correlation between 18F-FDG uptake at 2 weeks and 6 weeks after treatment, meaning patients who are effective in the second week after targeted therapy are usually effective in the sixth week after treatment. This study indicates that 18F-FDG PET imaging can predict the efficacy of targeted therapy at the early stage, without waiting for the middle stage or the end of treatment. Therefore, 18F-FDG PET imaging has been included in future studies as an essential biological detection method, providing a reference for the clinical decision of NAC and endocrine therapy. However, the specific threshold value [50] and the definition of good histopathologic response varies [51], and the optimal timing of interim PET is unclear [52]. These disparities need to be standardized.”

Reference:

  1. Chen X, Lu B, and Yan CG. Reproducibility of R-fMRI metrics on the impact of different strategies for multiple comparison correction and sample sizes. Hum Brain Mapp. 2018 39, 300-318.
  2. Pediconi F, Catalano C, Roselli A, Dominelli V, Cagioli S, Karatasiou A, Pronio A, Kirchin MA, and Passariello R. The challenge of imaging dense breast parenchyma: is magnetic resonance mammography the technique of choice? A comparative study with x-ray mammography and whole-breast ultrasound. Invest Radiol. 2009 44, 412-21.
  3. Whisenant JG, Romanoff J, Rahbar H, Kitsch AE, Harvey SM, Moy L, DeMartini WB, Dogan BE, Yang WT, Wang LC, Joe BN, Wilmes LJ, Hylton NM, Oh KY, Tudorica LA, Neal CH, Malyarenko DI, McDonald ES, Comstock CE, Yankeelov TE, Chenevert TL, and Partridge SC. Factors Affecting Image Quality and Lesion Evaluability in Breast Diffusion-weighted MRI: Observations from the ECOG-ACRIN Cancer Research Group Multisite Trial (A6702). J Breast Imaging. 2021 3, 44-56.
  4. Kim JY, Seo HB, Park S, Moon JI, Lee JW, Lee NK, Lee SW, and Bae YT. Early-stage invasive ductal carcinoma: Association of tumor apparent diffusion coefficient values with axillary lymph node metastasis. Eur J Radiol. 2015 84, 2137-43.
  5. Zhang M, Lian R, Zhang R, Hong Y, Feng W, and Feng S. The value of different imaging methods in the diagnosis of breast cancer: A protocol for network meta-analysis of diagnostic test accuracy. Medicine (Baltimore). 2021 100, e25803.
  6. Kim K, Shim SR, and Kim SJ. Diagnostic Values of 8 Different Imaging Modalities for Preoperative Detection of Axillary Lymph Node Metastasis of Breast Cancer: A Bayesian Network Meta-analysis. Am J Clin Oncol. 2021 44, 331-339.
  7. Zhang X, Liu Y, Luo H, and Zhang J. PET/CT and MRI for Identifying Axillary Lymph Node Metastases in Breast Cancer Patients: Systematic Review and Meta-Analysis. J Magn Reson Imaging. 2020 52, 1840-1851.

Round 2

Reviewer 2 Report

  1. the abstract includes some unclear sentences, in particular from line 22 to 25 Moreover this section should be more representative of article content.
  2. the new introduction is more focused on radionulcide imeging. However many parts are not clear because they sounds generic and not appropriate to the context.
  3. In line 40 , MRI, contrast imaging and US are not considered able to detect vascular and elastic tumour variations in breast cancer.
  4. some sentences, in ex.in line 43, are so less specific for breast tumour that they are appropriate for every kind of neoplasm.
  5. the sentence in line 85, concerning the warburg effect is quite generic.
  6. The section on FDG-PET should be rewrote; the authors have to put in evidence clear indications in breast canncer patients.
